# Microbial synthesis of Prussian blue for potentiating checkpoint blockade immunotherapy

Dongdong Wang [1,2,5], Jiawei Liu[1,3,5], Changlai Wang[2], Weiyun Zhang[4], Guangbao Yang[1], Yun Chen[1], Xiaodong Zhang [1], Yinglong Wu[1], Long Gu [1], Hongzhong Chen [1], Wei Yuan[1], Xiaokai Chen[1], Guofeng Liu[1], Bin Gao[1], Qianwang Chen [2] & Yanli Zhao [1] ✉

Cancer immunotherapy is revolutionizing oncology. The marriage of nanotechnology and immunotherapy offers a great opportunity to amplify antitumor immune response in a safe and effective manner. Here, electrochemically active *Shewanella oneidensis* MR-1 can be applied to produce FDA-approved Prussian blue nanoparticles on a large-scale. We present a mitochondria-targeting nanoplatform, MiBaMc, which consists of Prussian blue decorated bacteria membrane fragments having further modifications with chlorin e6 and triphenylphosphine. We find that MiBaMc specifically targets mitochondria and induces amplified photo-damages and immunogenic cell death of tumor cells under light irradiation. The released tumor antigens subsequently promote the maturation of dendritic cells in tumor-draining lymph nodes, eliciting T cell-mediated immune response. In two tumor-bearing mouse models using female mice, MiBaMc triggered phototherapy synergizes with anti-PDL1 blocking antibody for enhanced tumor inhibition. Collectively, the present study demonstrates biological precipitation synthetic strategy of targeted nanoparticles holds great potential for the preparation of microbial membrane-based nanoplatforms to boost antitumor immunity.

Cancer immunotherapy has been revolutionizing oncology and demonstrating varying degrees of success in certain solid cancers[1–3]. For instance, immune checkpoint blockade (ICB) can be applied to durably eliminate tumors by inhibiting intrinsic down-regulators of immunity, thus boosting the therapeutic efficacy[4]. However, clinical data suggest that only a small fraction (10–30%) of patients respond to the ICB[5]. Cancer nanomedicine, in combination with immunotherapy, offers great potential to amplify antitumor immune responses and sensitize tumors to immunotherapy in a safe and effective manner[6].

For instance, photo-based hyperthermia and reactive oxygen species (ROS) have been reported to induce immunogenic cell death for enhanced tumor immunogenicity[7,8]. The natural world provides a host of materials and inspiration for the field of nanomedicine. Recently, biological membrane-involved nanotechnology has been widely used in fabricating nanoplatforms for drug delivery, immune manipulation, and cancer treatment[9–11]. Although with promises, there are still some challenges, such as complicated procedures and large-scale manufacturing, in terms of clinical translation.

[1]School of Chemistry, Chemical Engineering and Biotechnology, Nanyang Technological University, Singapore 637371, Singapore. [2]Hefei National Research Center for Physical Sciences at the Microscale, University of Science and Technology of China, 230026 Hefei, P.R. China. [3]The Institute of Geology and Geophysics, Chinese Academy of Sciences, 100029 Beijing, P.R. China. [4]School of Biomedical Engineering, Shenzhen University, 518060 Shenzhen, P.R. China. [5]These authors contributed equally: Dongdong Wang, Jiawei Liu. ✉e-mail: zhaoyanli@ntu.edu.sg

Prussian blue analogs (PBAs) are a class of microporous metal-organic frameworks (MOFs)[12]. The parent Prussian blue (PB) is mixed-valence cyanide of iron in its $Fe^{II}$ and $Fe^{III}$ oxidation states with a formula of $Fe^{III}_4[Fe^{II}(CN)_6]_3 \cdot H_2O$. Since the inception of Diesbach in 1706, PB has been well-studied and widely used as a pigment. In 2003, PB capsule, namely Radiogardase, was approved by U.S. Food and Drug Administration (FDA) for the treatment of radioactive exposure (https://www.fda.gov/drugs/bioterrorism-and-drug-preparedness/fda-approves-first-new-drug-application-treatment-radiation-contamination-due-cesium-or-thallium). Recent studies also showed its superior photothermal conversion ability as a promising preclinical photothermal agent[13,14]. Generally, two chemical strategies are used for PB synthesis. One is a single-precursor strategy using either $K_4[Fe^{II}(CN)_6]$ or $K_3[Fe^{III}(CN)_6]$, and the other one is a double-precursor strategy using equivalently mixed $[Fe^{II}(CN)_6]^{4-}/[Fe^{III}(CN)_6]^{3-}$ and $Fe^{II}/Fe^{II}$ solution[15,16]. However, chemical strategies require relatively high reaction temperatures, harsh acidic environments, and additive surfactants to control the growth of PB nanocrystals and usually show low yield efficiency. Moreover, chemical strategies impose certain limitations for large-scale production due to the involved dynamics, thermodynamics, and safety concerns. Thus, it is highly desired to develop facile and eco-friendly strategies for controllable and large-scale synthesis of PB.

Microbe–mineral interactions are fundamental to global biogeochemical processes, including nitrogen and carbon cycles[17]. Geochemical evidence also indicated that $Fe^{III}$ might be the first external electron acceptor of global significance in microbial metabolism[18,19]. In particular, microorganisms with extracellular electron transfer capacity have been explored for biomining, bioremediation, and the production of bioenergy and biofuel[20–23]. Electrochemically active microorganisms such as *Geobacter metallireducens* GS-15 and *Shewanella oneidensis* MR-1 (*S. oneidensis* MR-1) can oxidize hydrogen or organic matter and transfer electrons to minerals containing $Fe^{III}$ or $Mn^{III}$ or $Mn^{IV}$ in the absence of respiratory terminal electron acceptors such as molecular oxygen. In addition, some microorganisms can induce the precipitation of nanomaterials such as noble metal nanoparticles, magnetite, and uraninite nanoparticles under mild conditions[24,25]. Given the inspiration behind the utilization of $Fe^{III}$ as an electron acceptor to support globally significant rates of respiration on early Earth, we thus speculated electrochemically active microorganisms could facilitate the biological reduction of $Fe^{III}$ into $Fe^{II}$, which subsequently coordinated with $[Fe^{III}(CN)_6]$ linkers to produce Prussian blue.

Here, we show that FDA-approved Prussian blue nanoparticles can be synthesized via a biological precipitation process on a large-scale taking advantage of the unique metabolic reduction capabilities of *S. oneidensis* MR-1 (Fig. 1a). During the self-respiration process, uniform PB MOFs are generated on the surface of *S. oneidensis* MR-1 bacteria (*S. oneidensis*-MOFs). The as-prepared *S. oneidensis*-MOFs are sonicated to form PB MOFs decorated bacteria membrane fragments (BaM) (Fig. 1b). To enhance the photo-induced damage, we construct a mitochondria-targeting therapeutic nanoplatform, namely MiBaMc, consisting of BaM with further surface modifications of ROS-generating chlorin e6 (Ce6) and mitochondria-targeting triphenylphosphine (TPP). After tumoral accumulation and endocytosis of MiBaMc, both hyperthermia, and ROS are generated to induce immunogenic cell death of cancer cells under external light irradiation (Fig. 1c). Furthermore, MiBaMc shows potent tumor inhibition ability in two tumor-bearing mouse models (immunogenic triple-negative breast cancer and colorectal cancer) when combined with clinically approved anti-PDL1 blocking antibody (aPDL1). Our study demonstrates that MiBaMc is a promising antitumor agent.

## Results
### Biological precipitation of Prussian blue
In this study, we chose the electrochemically active bacterium *S. oneidensis* MR-1 as a model due to its capacity to use minerals that contain $Fe^{III}$ as terminal electron acceptors[26,27]. As shown in Fig. 1b, the membrane protein complex (MtrA, MtrB, and MtrC) continuously furnish electrons from the periplasmic proteins to the bacterial surfaces[28,29]. The biological precipitation of Prussian blue MOFs by *S. oneidensis* was initiated by adding ferric citrate and $K_3[Fe^{III}(CN)_6]$ to an early exponential growth culture. Transmission electron microscopy (TEM) and scanning electron microscopy (SEM) images indicated the successful precipitation of PB MOFs onto the surface of *S. oneidensis* MR-1 (*S. oneidensis*-MOFs) (Fig. 2a, b and Supplementary Fig. 1). The calculated yield efficiency of PB MOFs under the microbial synthesis strategy was 43.4%, which is higher than one-pot chemical synthesis strategy with a value of 21.6%. After intensive sonication for 2 h, PB MOFs decorated bacteria membrane fragments (BaM) were obtained (Fig. 2c and Supplementary Fig. 2). The weight ratio of PB and bacteria membrane in BaM was calculated to be 2.06/7.94. There were no noticeable changes in the morphology after further modifications of mitochondria-targeting TPP and ROS-generating Ce6 (Fig. 2d). The weight ratio of PB, Ce6, TPP, and bacteria membrane in MiBaMc was calculated to be 2.06/2.23/1.38/7.94. High-angle annular dark-field scanning transmission electron microscopy (HAADF-STEM) and energy-dispersive X-ray spectroscopy (EDS) mapping images revealed the existence of brighter parts composed of Fe, N, and C elements, which is consistent with the components of Prussian blue (Fig. 2e). High-resolution STEM image revealed an interplanar spacing of 5.13 Å, which is consistent with the calculated lattice distance of Prussian blue (Fig. 2f and inset). The selected area electron diffraction (SEAD) pattern from the ⟨100⟩ zone axis showed the semirings correspond with the main lattice planes of Prussian blue (Fig. 2g, h). The above results indicated the successful synthesis of Prussian blue using *S. oneidensis* MR-1. Powder X-ray diffraction (PXRD) patterns (Fig. 2i) further confirmed the successful biological precipitation of Prussian blue with the main diffraction peaks[30]. Additionally, the grazing-incidence wide-angle X-ray scattering (GIWAXS) pattern showed strong signals from Prussian blue nanocrystal, which is consistent with the PXRD result (Fig. 2j). The synthesis of the *S. oneidensis*–Prussian blue hybrid could be scaled largely by increasing the initial volume of the culture medium (Fig. 2k). The dry tablet of BaM after freeze-drying showed the long-term storage possibility.

The electrochemical analysis was conducted to reveal the mechanism of this biosynthesis process. As shown in Fig. 2l, the reduction potential of *S. oneidensis* MR-1 was around −0.23 V vs. RHE (Supplementary Fig. 3). Long-term chronoamperometric measurement at −0.26 V vs. RHE was carried out to simulate the reduction ability of the bacteria. The current showed no changes after adding $K_3[Fe^{III}(CN)_6]$ solution, indicating that *S. oneidensis* MR-1 was unable to reduce the central coordinated $Fe^{III}$ ions in $K_3[Fe^{III}(CN)_6]$. However, with the further addition of ferric citrate, the current increased dramatically, suggesting the bacteria can reduce free $Fe^{III}$ ions from ferric citrate. The selective reduction ability of bacteria ensured the presence of both $Fe^{II}$ and $[Fe^{III}(CN)_6]$ linkers in the system. UV–vis–NIR spectra identified the successful conjugation of Ce6 with a characteristic peak centered at 671 nm (Fig. 2m). Besides, MiBaMc showed a typical absorption peak ranging from 600 to 900 nm of Prussian blue[31]. Dynamic light scattering (DLS) tests showed an average size of 120 nm, which is suitable for biological applications (Supplementary Fig. 4). The photothermal effects of MiBaMc were studied under irradiation of 808 nm laser, showing a time-dependent temperature increase within 10 min with a comparable photothermal effect to Prussian blue (Fig. 2n, o).

### Cellular uptake and immunostimulation of MiBaMc
The fluorescence quantum yield efficiency of free Ce6, BaMc, and MiBaMc in PBS was calculated to be 0.17, 0.16, and 0.163, respectively. Confocal laser scanning microscopy (CLSM) images showed a time-dependent internalization enhancement of the MiBaMc in murine

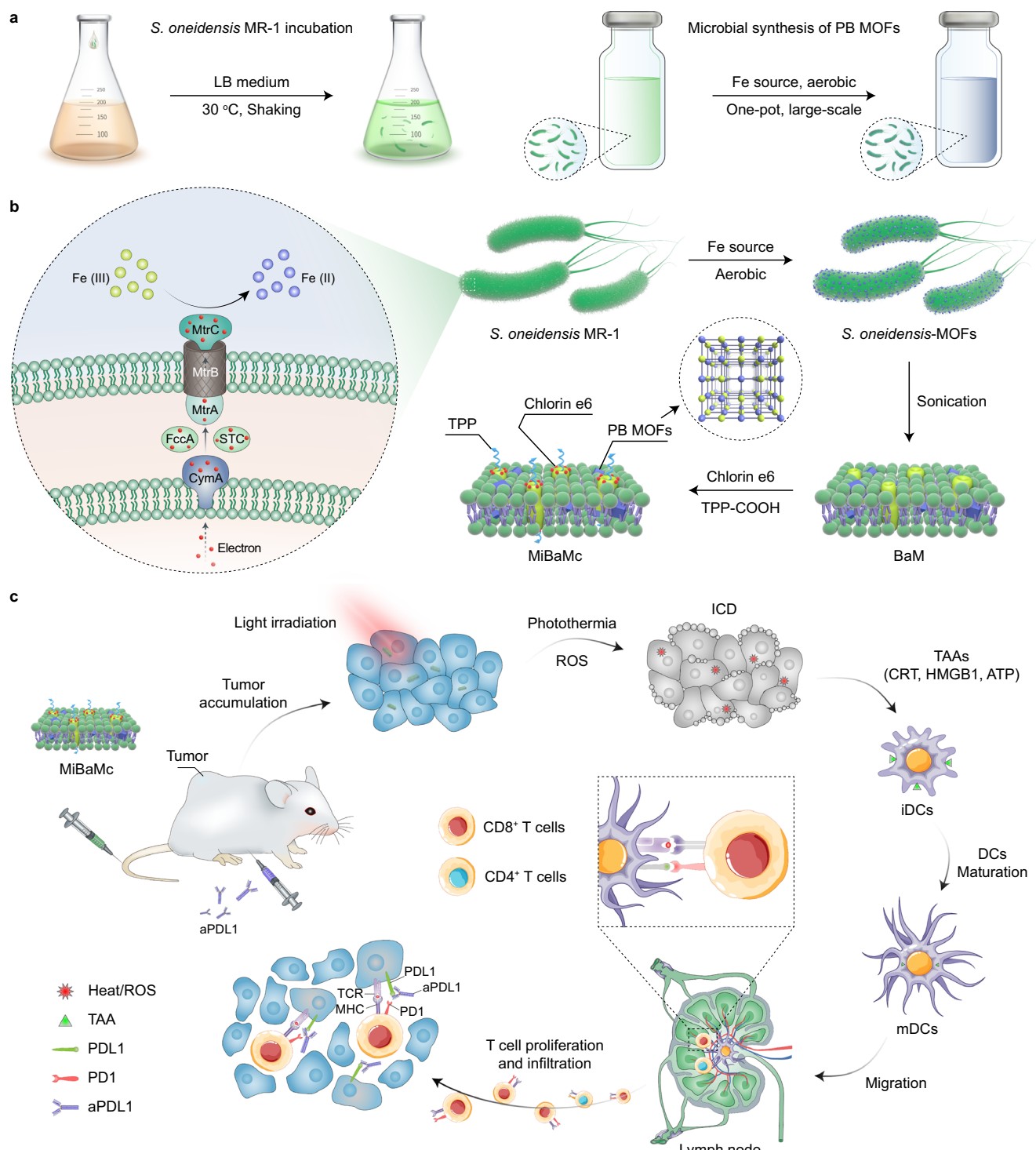

**Fig. 1 | Microbial synthesis of MiBaMc combined with aPDL1 antibody for enhanced tumor immunotherapy. a** Schematic illustration of the incubation of *S. oneidensis* MR-1 and subsequently one-pot large-scale microbial synthesis of FDA-approved Prussian blue MOFs. **b** Schematic illustration of the biological pre-cipitation of PB MOFs coated *S. oneidensis* MR-1 hybrid (*S. oneidensis*-MOFs) through extracellular electron transfer pathways, in which extracellular electron flux via the MtrCAB pathway in *S. oneidensis* MR-1 can reduce Fe$^{III}$ into Fe$^{II}$ for the following generation of PB MOFs. The PB MOF decorated bacteria membrane ragments (BaM) were obtained after the sonication of *S. oneidensis*-MOFs. BaM was further modified with mitochondria-targeting agent TPP and photosensitizer Ce6 to generate MiBaMc with mitochondria-targeting ability. **c** Schematic illustration of the mitochondria-targeting MiBaMc system-induced ICD combined with aPDL1 for enhanced tumor immunotherapy. ICD immunogenic cell death, TAA tumor-associated antigen, CRT calreticulin, HMGB1 high-mobility group protein box 1, DCs dendritic cells, iDCs immature DCs, mDCs mature DCs, TCR T cell receptor, MHC major histocompatibility complex.

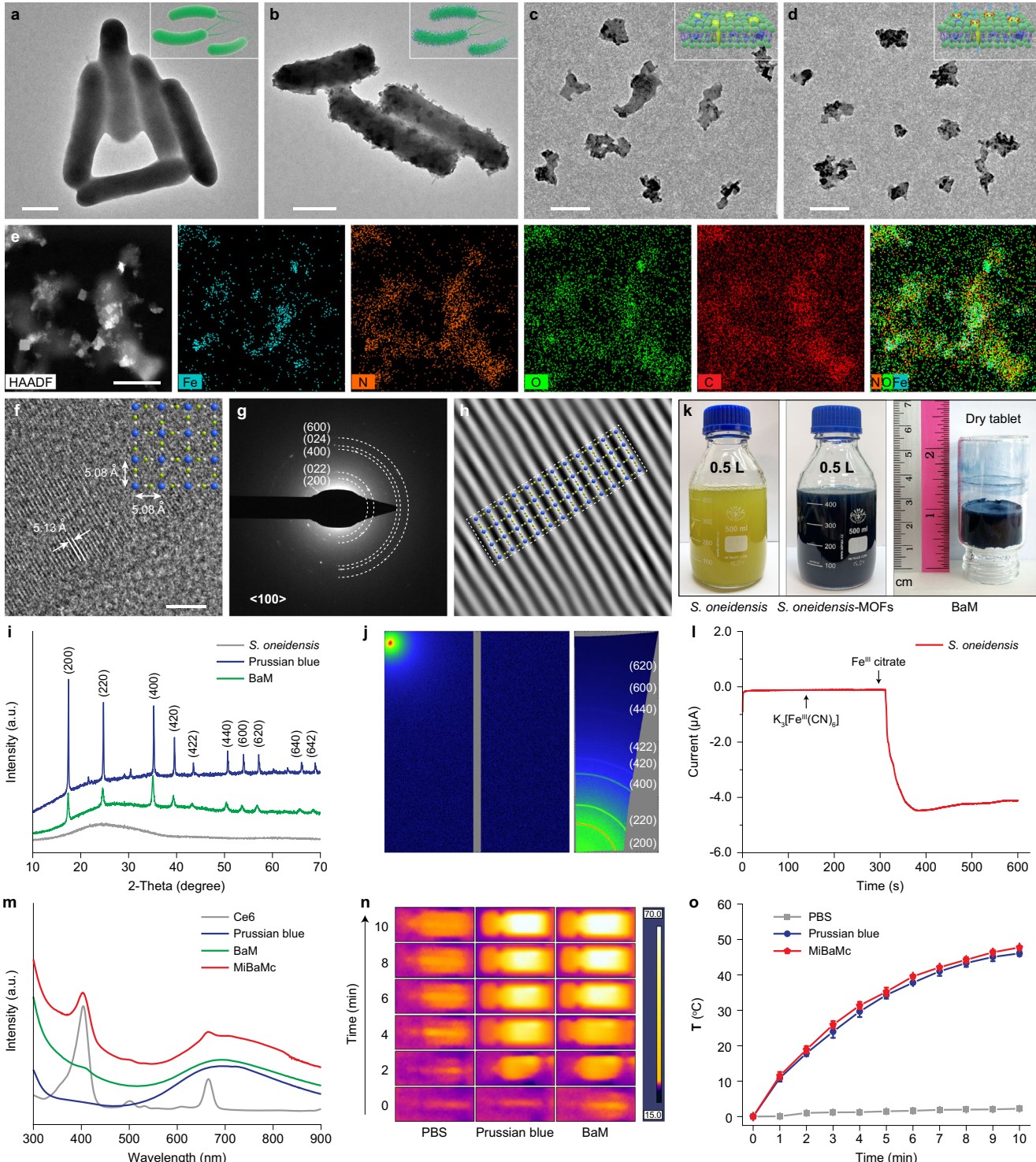

**Fig. 2 | Synthesis and morphology characterizations.** TEM images of **a** *S. oneidensis* MR-1 and **b** *S. oneidensis*-MOFs. Scale bar, 500 nm. TEM images of **c** BaM and **d** MiBaMc. Scale bar, 200 nm. **e** HAADF-STEM image of MiBaMc and corresponding EDS elemental mapping. Scale bar, 200 nm. **f** High-resolution TEM image of MiBaMc. Scale bar, 5 nm. The experiments in **a–j** were repeated three times with similar results. **g** SEAD pattern of PB from the 100⟩ zone. **h** Corresponding lattice distance of synthesized PB MOFs. **i** XRD patterns of *S. oneidensis* MR-1, BaM, and PB MOFs. **j** GIWAXS pattern of BaM. **k** Large-scale synthesis of *S. oneidensis*-MOFs with 0.5 L culture medium and the dry tablet of BaM after dehydration treatment.

**l** Potentiostatic ferric citrate reduction with a working electrode of *S. oneidensis* MR-1 coated on GC and held at a potential of 0.20 V versus Ag/AgCl. $K_3[Fe^{III}(CN)_6]$ (0.1 M) was added to the buffer at $t = 150$ s, and ferric citrate was added to the buffer at $t = 300$ s. **m** UV–vis–NIR spectra of PB, free Ce6, BaM, and MiBaMc. **n** Thermal photograph of PBS, Prussian blue, and BaM under irradiation of 808 nm laser (0.5 W cm$^{-2}$). The experiments in **l–n** were repeated three times with similar results. **o** Photothermal curves of PBS, Prussian blue, and MiBaMc at a PB concentration of 100 µg mL$^{-1}$ under 808 nm laser irradiation ($n = 3$ independent experiments). Data are presented as mean values ± SD. Source data are provided as a Source Data file.

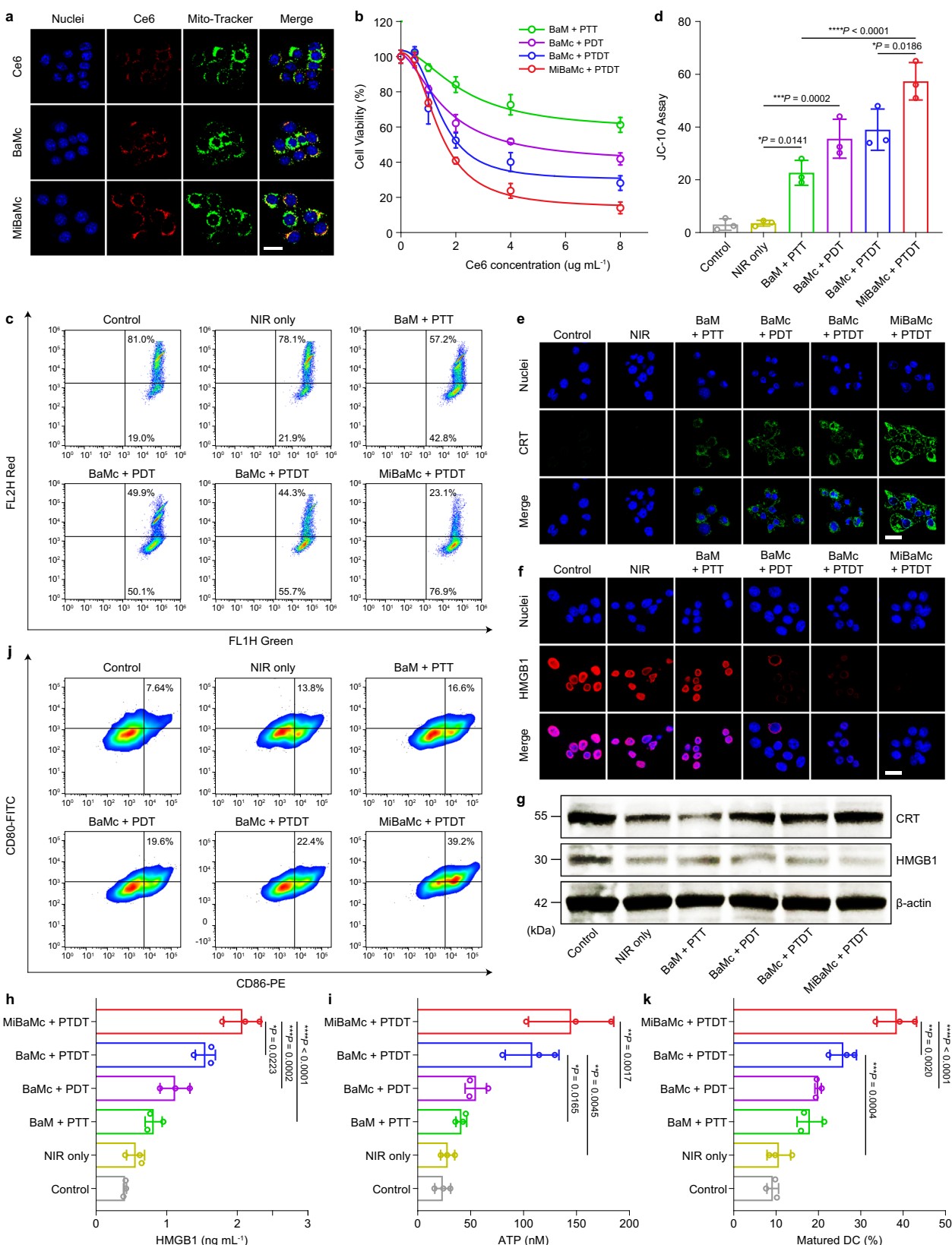

breast cancer (4T1) cells (Supplementary Fig. 5). Moreover, the Pearson's colocalization coefficients between the red Ce6 channel and the green mitochondria channel for free Ce6, BaMc, and MiBaMc were calculated to be $0.18 \pm 0.08$, $0.29 \pm 0.10$, and $0.51 \pm 0.15$, respectively. (Fig. 3a). The above results demonstrated the mitochondria-targeting ability of MiBaMc. The dark cytotoxicity of each formulation was also

evaluated via the standard MTT assay. Results showed favorable biocompatibility of all formulations under dark conditions within a wide range of concentrations (Supplementary Fig. 6). Under irradiation, each formulation displayed a concentration-dependent killing efficiency against 4T1 cancer cells (Fig. 3b). Remarkably, the combined photothermal-photodynamic therapy (PTDT) of MiBaMc showed a

**Fig. 3 | Uptake of MiBaMc and ICD in tumor cells triggered by MiBaMc. a** CLSM images showing the intracellular uptake of free Ce6, BaMc, and MiBaMc in 4T1 cancer cells after 8 h incubation. Blue, nucleus stained with Hoechst 33342; red, Ce6 fluorescence; green, mitochondria stained with Mito-Tracker green. Scale bar, 25 μm. The experiments in **a** were repeated three times with similar results. **b** Relative viability of 4T1 cancer cells after treatment with different formulations ($n = 4$ independent experiments). **c** Flow cytometry analysis of the mitochondrial membrane potential of 4T1 cells after treatment with different formulations. **d** Corresponding green fluorescence intensity of different groups based on the flow cytometry results in **c** ($n = 3$ independent experiments). Statistical analysis was conducted by one-way ANOVA with Tukey's tests. $^*P < 0.05$, $^{***}P < 0.001$, $^{****}P < 0.0001$. CLSM images showing **e** CRT exposure and **f** HMGB1 release in 4T1 cells after each treatment. Blue, nucleus stained with Hoechst 33342; Green, CRT stained with mouse anti-CRT primary antibody and then with Alexa Flour-488 conjugated goat anti-mouse secondary antibody; Red, HMGB1 stained with rabbit anti-HMGB1 primary antibody and then with Alexa Flour-647 conjugated goat anti-rabbit secondary antibody. Scale bar, 25 μm. **g** Western blot assay of HMBG1 and CRT proteins expressed in 4T1 cells after treatment with each formulation. The experiments in **e-g** were repeated three times with similar results. Quantification of extracellular release of **h** HMGB1 and **i** ATP in culture medium after different treatments ($n = 3$ independent experiments). Statistical analysis was conducted with ANOVA with Tukey's tests. $^*P < 0.05$, $^{**}P < 0.01$, $^{***}P < 0.001$, $^{****}P < 0.0001$. **j** Flow cytometry analysis of the matured DCs (CD80$^+$CD86$^+$) after different treatments ($n = 3$). **k** Quantification of the percentage of matured DCs after different treatments ($n = 3$ independent experiments). Statistical analysis was conducted with ANOVA with Tukey's tests. $^{**}P < 0.01$, $^{***}P < 0.001$, $^{****}P < 0.0001$. Data are presented as mean values ± SD. Source data are provided as a Source Data file.

synergistic effect with an 86.0% killing efficiency, much higher than BaMc + PTDT, BaMc + PDT, and BaM + PTT groups on account of the mitochondria targeting ability.

On account of the key role of mitochondria in apoptosis, we speculated that cell death could be initiated by the amplification of oxidative stress in mitochondria[32,33]. The intracellular ROS levels were studied with 2,7-dichlorofluorescein-diacetate (DCFH-DA) as a fluorescent probe[34]. Boosted ROS generation was observed in the MiBaMc group compared to other groups (Supplementary Fig. 7). It is well known that mitochondrial membrane potential (MMP) is a critical parameter in evaluating cell death[35]. A mitochondrial-specific dual-fluorescence probe (JC-10 dye) was applied to study MMP (Fig. 3c, d). Results showed a prominent JC-10 fluorescence shift from red to green for the MiBaMc group, indicating the depolarization and decrease in MMP, the early stage of cell apoptosis[36]. CLSM images also revealed the same tendency (Supplementary Fig. 8). Flow cytometry and dead–live staining results suggested the highest apoptosis level in the MiBaMc group (Supplementary Figs. 9 and 10). Overall, MiBaMc could effectively induce amplified apoptosis via its mitochondrial-targeting ability.

Treatment with different formulations including BaM + PTT, BaMc + PDT, BaMc + PTDT, and MiBaMc + PTDT, obviously induced calreticulin (CRT) exposure in cancer cells compared with groups treated with phosphate-buffered saline (PBS, control) and NIR irradiation (Fig. 3e). At the same time, the MiBaMc group showed the most prominent extracellular release of high-mobility group protein box 1 (HMGB1) (Fig. 3f). Furthermore, western blot was performed to evaluate the expression of CRT and HMGB1 (Fig. 3g and Supplementary Fig. 11). Results showed significantly increased expression of CRT and reduced expression of HMGB1 in 4T1 cells after treatment with MiBaMc, which corroborated the CLSM results. The extracellular release of HMGB1 and ATP in the supernatant of 4T1 cancer cells was also evaluated through ELISA analysis (Fig. 3h, i). The MiBaMc-treated group showed the highest HMGB1 secretion and ATP release levels compared with the other groups. Furthermore, the maturation of DCs after treatment of each formulation was also studied by flow cytometry (Fig. 3j, k and Supplementary Figs. 12 and 13). The percentage of matured DCs of the MiBaMc-treated group increased by ~3.21, ~1.85, and ~1.32-fold relative to those in the BaM + PTT, BaMc + PDT, and BaMc + PTDT groups, respectively. Overall, those data demonstrated that MiBaMc could effectively induce ICD and DC maturation for enhanced antigen presentation.

### In vivo biocompatibility and tumor accumulation of MiBaMc

To identify the optimal therapeutic windows, 4T1 tumor-bearing mice were intravenously administered with free Ce6 and MiBaMc at an equivalent Ce6 concentration of 4 mg kg$^{-1}$. Fluorescence imaging was monitored at pre-determined time intervals to determine their intratumoral accumulation (Supplementary Fig. 14). For MiBaMc-treated mice, fluorescence signal within the tumor region reached its maximum value at 12 h post-administration and maintained a relatively high fluorescence even for 24 h, suggesting the favorable accumulation of MiBaMc. Ex vivo imaging at 24 h post-administration showed a remarkable tumor accumulation within the tumor region compared with other tissues such as the liver, spleen, lung, kidney, and heart, further suggesting the favorable tumor accumulation. Furthermore, the ICP test also indicated the high accumulation in tumor tissue by testing the Fe amount (Supplementary Fig. 15). The preferential tumor accumulation of MiBaMc suggested the passive tumor-targeting ability owing to the enhanced permeation and retention (EPR) effect as well as the inherent mechanical softness of biological membrane[37–39]. Besides, there was no detectable hemolysis of red blood cells after incubation with MiBaMc even with a concentration as high as 1000 μg mL$^{-1}$ (Supplementary Fig. 16).

### In vivo tumor inhibition of MiBaMc

Encouraged by the above results, in vivo therapy was subsequently performed to assess the ability of MiBaMc to suppress tumor growth in a 4T1 tumor model inoculated in the right flank of Balb/c mice on day 0. Thermal imaging showed the temperature of 4T1 tumor-bearing tumor tissue increased ~44.3 °C 12 h post-administration of MiBaMc under continuous 808 nm laser irradiation for 10 min, suggesting the favorable photothermal conversion ability in vivo (Supplementary Fig. 17). As indicated in the therapeutic schedule (Fig. 4a), on day 7, 8, and 9, the animals from aPDL1 (group 2), BaMc + PTDT + aPDL1 (group 8), and MiBaMc + PTDT + aPDL1 (group 9) were treated by intraperitoneal administration of aPDL1 IgG with a daily amount of 50 μg. On day 10 post tumor inoculation, the animals from 9 groups were intravenously administrated with each formulation. Based on the in vivo fluorescence imaging results, NIR irradiation was applied 12 h post-administration of each formulation.

Tumor growth curves were monitored for 18 days. Compared with the PBS group, aPDL1, free Ce6, BaMc + PDT, and BaMc + PTT monotherapy exhibited limited tumor growth suppressive ability. The average tumor volume in those groups reached 550 mm$^3$ within 28 days (Fig. 4b, c and Supplementary Fig. 18). In contrast, BaMc + PTDT, MiBaMc + PTDT, and BaMc + PTDT + aPDL1 showed enhanced tumor inhibition, with an average tumor volume of 480.7, 396.6, and 181.1 mm$^3$, respectively. Remarkably, MiBaMc + PTDT + aPDL1 achieved 95.5% reduction in tumor growth compared to the PBS group. Consistently, the average weight of tumors in the MiBaMc + PTDT + aPDL1 group was also the lowest among all treated groups (Fig. 4d). Additionally, no apparent body weight variation was observed in treatment groups, further confirming the favorable biocompatibility of all formulations (Fig. 4e). Furthermore, the antitumor activity was performed on a more immunosuppressive MC38 colorectal tumor model. Results showed a similar tendency under the same treatment dose (Fig. 4c, f–h and Supplementary Fig. 19). Administration of aPDL1 failed to suppress the tumor growth. In contrast, MiBaMc + PTDT + aPDL1

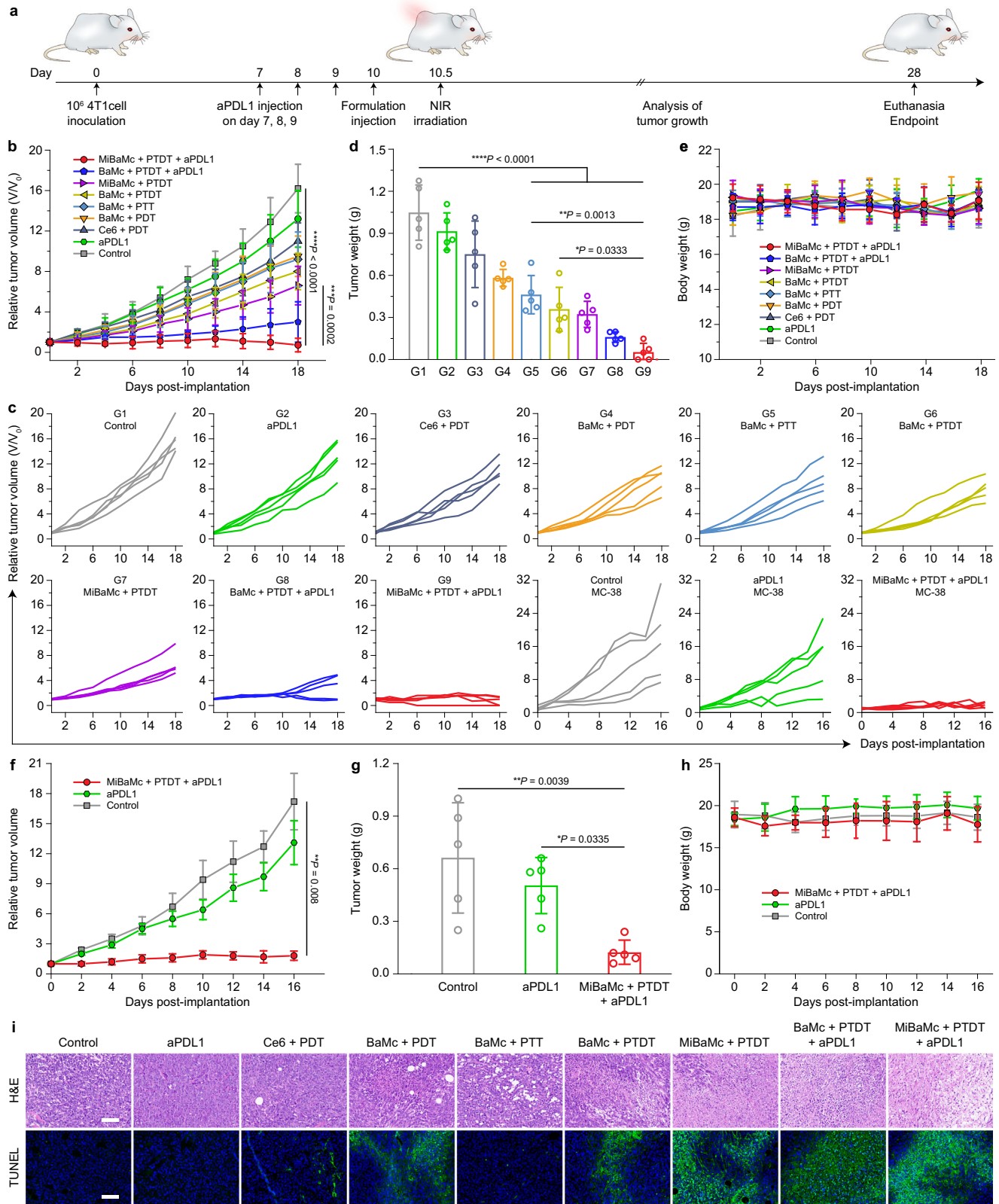

group showed a prominent suppression of MC38 tumor growth with an inhibition rate of 89.4%.

Immunohistochemical staining was further applied to support the antitumor effect. 4T1 tumor tissues were harvested after different treatments, and tumor sections were stained and imaged (Fig. 4i). Haematoxylin and eosin (H&E) results showed the most

prominent damage in MiBaMc + PTDT + aPDL1 group. The TUNEL (terminal deoxynucleotidyl-transferase-mediated nick end labeling) signal also significantly increased in MiBaMc + PTDT + aPDL1 group. Those results demonstrated the enhanced tumor cell death in MiBaMc-treated mice, consistent with the enhanced antitumor activity of the MiBaMc + PTDT + aPDL1 group.

**Fig. 4 | In vivo antitumor activity of mice bearing 4T1 and MC38 tumors.**
**a** Schematic illustration of the treatment schedule. On day 7 post tumor implantation, mice were randomly divided into 9 groups. Groups 2, 8, and 9 received aPDL1 every day at a daily dose of 40 µg per mouse through intraperitoneal injection for 3 days. Then mice were treated with each formulation through intravenous injection and light irradiation 12 h post-administration. **b** 4T1 tumor volumes of different groups were measured every 2 days ($n = 5$ mice). Statistical analysis was conducted by a two-tailed unpaired Student's $t$-test. $^{**}P < 0.01$, $^{****}P < 0.0001$. **c** Individual 4T1 and MC38 tumor growth curves during treatment. **d** Average 4T1 tumor weight of different groups at the endpoint of treatment ($n = 5$ mice). Statistical analysis was conducted with ANOVA with Tukey's tests. $^{*}P < 0.05$, $^{**}P < 0.01$, $^{****}P < 0.0001$. **e** Average body weight of 4T1 tumor-bearing mice of different groups during the treatment ($n = 5$ mice). **f** MC38 tumor volumes of different groups were measured every 2 days ($n = 5$ mice). Statistical analysis was conducted by a two-tailed unpaired Student's $t$-test. $^{*}P < 0.05$. **g** Average MC38 tumor weight of different groups at the endpoint of treatment ($n = 5$ mice). Statistical analysis was conducted by one-way ANOVA with Tukey's tests. $^{*}P < 0.05$, $^{**}P < 0.01$. **h** Average body weight of MC38 tumor-bearing mice during the treatment ($n = 5$ mice). **i** Representative histological analysis and immunofluorescence images of 4T1 tumor tissues stained by H&E and TUNEL assay after different treatments ($n = 3$ mice). Scale bar, 100 µm. The experiments in **i** were repeated three times with similar results. Data are presented as mean values ± SD. Source data are provided as a Source Data file.

## In vivo biocompatibility

The biosafety of MiBaMc was fully studied. Liver and kidney function analysis of mice treated with MiBaMc for 7 and 15 days showed no difference as compared with the control group, demonstrating its long-term biosafety and ensuring its potential for clinical translation (Supplementary Fig. 20). The blood routine analysis exhibited a slight increase in the number of WBC, lymphocyte, monocytes, and neutrophils on day 1 post-administration. However, the levels of these indicators gradually restored to the normal range on day 3 and day 7 post-administration (Supplementary Figs. 21–24). These results demonstrated that MiBaMc could induce a low-level immune response without causing long-term harm. Furthermore, H&E staining on main organs (heart, liver, spleen, lung, kidney, and intestine) showed no detectable acute pathological changes after different treatments (Supplementary Fig. 25). Overall, these results demonstrated the high intratumoral accumulation and favorable biocompatibility of MiBaMc, supporting its enormous potential for further clinical translation.

## In vivo mechanistic studies of photo-immunotherapy

To investigate the immune response after administration of each formulation, DCs activation in tumor-draining lymph nodes was studied. Flow cytometry results observed an enhanced DCs maturation (26.2% CD86+CD80+ DCs) in MiBaMc + PTDT + aPDL1 treated group, which was 2.00-, 1.42-, and 1.34-fold higher than the control group (13.1%), aPDL1 group (18.5%), and BaMc + PTDT group (19.5%), respectively (Fig. 5a, b and Supplementary Fig. 26). Both cytotoxic T lymphocytes (CTL) (CD3+CD8+ T cells) and helper T cells (CD3+CD4+ T cells) are critical to regulating adaptive immunities. As shown in Fig. 5c,d, the percentage of activated CD3+CD8+ T cells in the MiBaMc + PTDT + aPDL1 group (29.1%) was 1.22-, 1.46-, and 2.75-fold higher than BaMc + PTDT + aPDL1, MiBaMc + PTDT, and control groups, respectively. In addition to CTL recruitment, MiBaMc triggered phototherapy combined with aPDL1 elicited the most significant infiltration of helper T cells (42.5%) in the tumor, showing 1.18-, 1.26-, and 2.31-fold increase in population compared to the BaMc + PTDT + aPDL1, MiBaMc + PTDT, and control groups, respectively (Fig. 5c,e). Consistently, the frequencies of both CD3+CD8+ and CD3+CD4+ cells in the spleen were also the most significantly elevated in MiBaMc + PTDT + aPDL1 group (Supplementary Fig. 27). Overall, the above results indicated that the MiBaMc triggered phototherapy combined with aPDL1 could effectively elicit the immune responders, an essential step to mount an antitumor immunity.

In addition, the secretions of proinflammatory cytokines such as interferon-γ (IFN-γ), tumor necrosis factor-α (TNF-α), and interleukin-6 (IL-6) in peripheral blood serum were analyzed with ELISA kits after different treatments (Fig. 5f–h). Consistent with the DCs maturation results, MiBaMc + PTDT + aPDL1 triggered the highest secretion of cytokines, further demonstrating the most effective ICD induced by BaMc + PTDT + aPDL1. For the MC38 tumor model, a similar tendency of the release of the cytokines could be observed (Supplementary Fig. 28). Furthermore, the expression of representative biomarkers of ICD (HMGB1 and CRT) within tumor sections was studied by immunofluorescence staining. Results showed significantly decreased HMGB1 expression within the nuclei and amplified CRT exposure in MiBaMc + PTDT + aPDL1 group (Fig. 5i-k). Besides, MiBaMc + PTDT + aPDL1 group showed the highest population of CD4+ helper T cells and CD8+ cytotoxic T cells within the tumor sections, as verified by the immunofluorescence staining (Fig. 5i and Supplementary Fig. 29). Afterward, the immunosuppressive tumor microenvironment was studied. The populations of Tregs (CD3+CD4+Foxp3+) for MiBaMc + PTDT + aPDL1 and MiBaMc + PTDT dramatically decreased by 84.3% and 63.4% compared to the control group injected with PBS, respectively (Supplementary Fig. 30). Collectively, those results demonstrated that MiBaMc triggered phototherapy combined with aPDL1 could effectively induce antitumor immune responses with both increased frequency of positive immune responders and decreased frequency of negative immune inhibitors, thus inhibiting the growth of the tumor.

## Proteomics analysis

A proteomics study was applied to accurately identify and quantify proteins involved during the treatment[40]. A total of 6392 proteins were identified. 92 differentially expressed proteins (70 up-regulated and 22 down-regulated) under a threshold with fold changes ≥ 1.5 and $p$ values < 0.05 were found in MiBaMc + PTDT + aPDL1 group compared to the control group as depicted in the heatmap (Fig. 6a). Based on the gene ontology (GO) analysis, we found that proteins associated with responses to stimulus, processes related to stimulus, immune system, signaling, cell killing, and antioxidant activity were more up-regulated in MiBaMc + PTDT + aPDL1 group relative to the control group (Fig. 6b). In contrast, proteins that related to cell integrity, cell growth, and structural molecular activity were more down-regulated. Those results indicated that MiBaMc induced strengthened tumor damage, which should be attributed to the synergistic effect induced by hyperthermia and ROS. Furthermore, prominent upregulation of proteins associated with antioxidant activity also directly indicated the enhanced oxidative pressure of tumors in the MiBaMc + PTDT + aPDL1 group. The Venn diagram shown in Fig. 6c revealed obvious overlaps between proteins associated with heat shock and cell death, as well as oxidative stress and cell death. Thus, both photothermal-induced hyperthermia and photodynamic-induced oxidative damage triggered by MiBaMc contributed to ICD. This phenomenon was further supported by the ROS staining of tumor sections (Supplementary Fig. 31). In addition, significant enhancement in the protein level associated with immune response was observed in the MiBaMc + PTDT + aPDL1 group (Fig. 6d), suggesting the antitumor capacity should be attributed to the systemic immune activation of MiBaMc triggered phototherapy in combination with aPDL1. Moreover, the Venn diagram also indicated prominent overlaps between proteins associated with immune response, inflammatory response, and bacterial infection, indicating that multiple immunological processes were activated and interacted to promote antitumor immunity. Herein, we deduced that while MiBaMc contains only the bacteria membrane, some proteins or membrane components (e.g., lipopolysaccharides, peptidoglycan, and

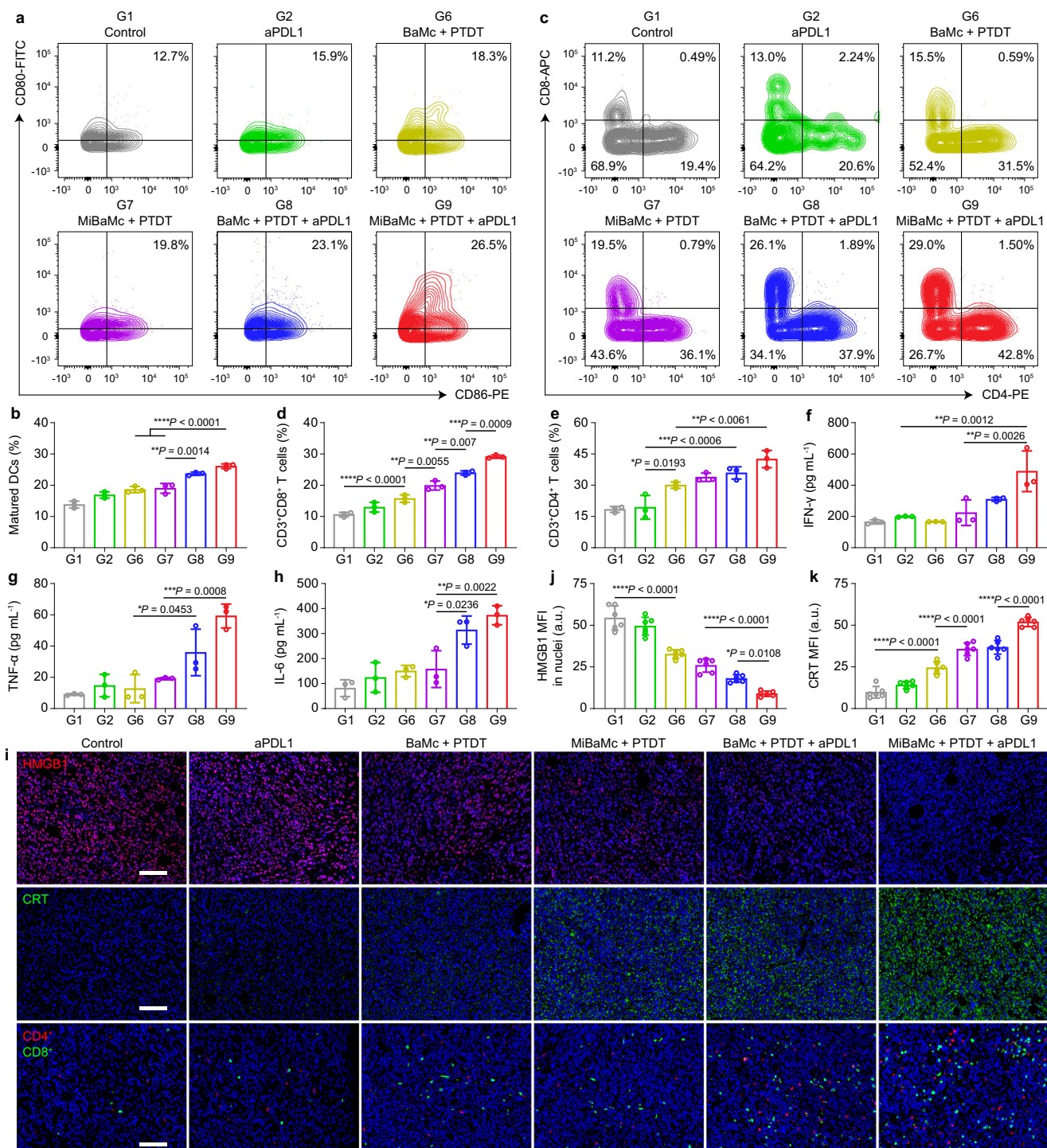

**Fig. 5 | Immuno-response of different treatment groups. a** Representative flow cytometric plots of the maturation of DCs in lymph nodes after treatment of each formulation ($n = 3$ mice). **b** Quantitative analysis of the matured DCs based on flow cytometric results in **a** ($n = 3$ mice). Statistical analysis was conducted by one-way ANOVA with Tukey's tests. $^{**}P < 0.01$, $^{****}P < 0.0001$. **c** Representative flow cytometric plots of the T cells in 4T1-bearing tumor tissue gating on CD3$^+$ cells after treatment of each formulation ($n = 3$ mice). **d**, **e** Quantitative analysis of the CD3$^+$CD8$^+$ cytotoxic T cells and CD3$^+$CD4$^+$ helper T cells as a percentage of CD3$^+$ lymphocytes based on flow cytometric results in **c** ($n = 3$ mice). Statistical analysis was conducted by one-way ANOVA with Tukey's tests. n.s. represents none of significance, $^*P < 0.05$, $^{**}P < 0.01$, $^{***}P < 0.001$, $^{****}P < 0.0001$. Quantitative analysis of cytokine

expression levels of **f** IFN-γ, **g** TNF-α, and **h** IL-6 in serum of 4T1 tumor-bearing mice after treatment of each formulation ($n = 3$ mice). Statistical analysis was conducted by one-way ANOVA with Tukey's tests. $^*P < 0.05$, $^{**}P < 0.01$, $^{***}P < 0.001$.
**i** Immunofluorescence staining of HMGB1 in nuclei (upper), CRT (middle), and CD4$^+$ and CD8$^+$ T cells (lower) in tumor sections after treatment of each formulation. Nuclei (blue) were stained by Hoechst 33342. Scale bar, 100 μm. Quantitative analysis of HMGB1 expression in **j** nuclei and **k** CRT expression of tumor sections after treatment with each formulation ($n = 6$ samples). Statistical analysis was conducted by one-way ANOVA with Tukey's tests. $^*P < 0.05$, $^{**}P < 0.01$, $^{***}P < 0.001$. Data are presented as mean values ± SD. Source data are provided as a Source Data file.

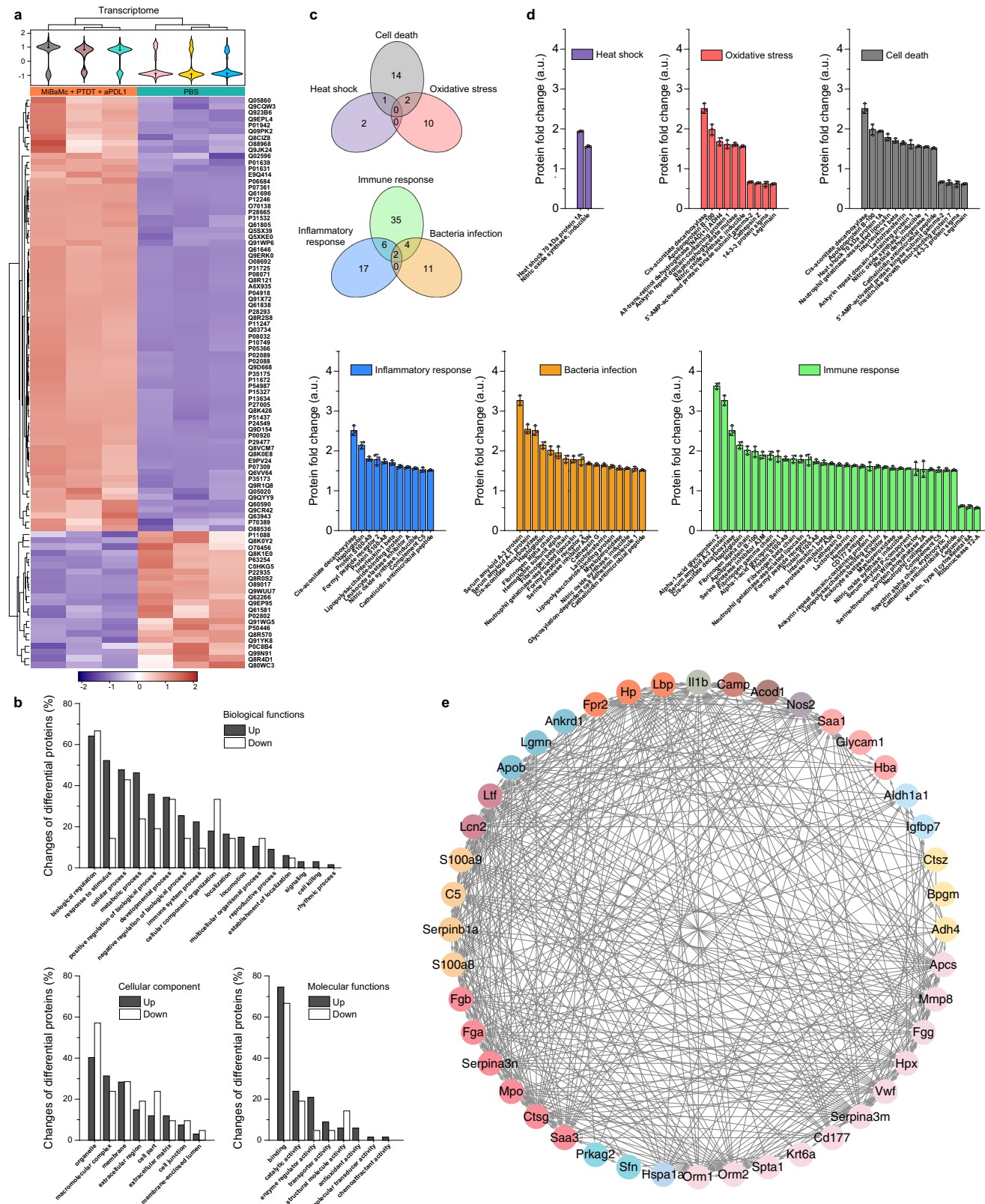

flagellum) still preserved their immunogenicity within the tumor site[41]. During this process, both hyperthermia and ROS triggered the release of tumor-associated antigens. The membrane component originating from bacteria might inherit some immunomodulatory features, thus enabling boosted systemic immune responses. Moreover, the tendency could also be drawn from the protein–protein interaction network (Fig. 6e and Supplementary Fig. 32). Overall, these proteomic data suggested that MiBaMc-triggered phototherapy combined with aPDL1 could promote multiple immunological processes to activate antitumor immunity.

## Discussion

Combining nanomedicine with immunotherapy is a highly desired cancer therapeutic modality due to its promising usage to achieve

**Fig. 6 | Quantitative proteomics analysis for exploring the antitumor mechanism of MiBaMc combined with aPDL1. a** Hierarchical clustering analysis of 92 differentially expressed proteins in tumor tissues after treatment with PBS and MiBaMc + PTDT + aPDL1 (*n* = 3 mice). The heatmap shows significantly upregulated (70) and downregulated (22) proteins (fold change ≥ 1.5 and *P*-value < 0.05). Fold change values were color-coded from purple (downregulation) to red (upregulation). **b** GO terms enrichment analysis of differentially expressed proteins. The abscissa shows enriched GO function classifications, namely three major categories: biological functions, cellular components, and molecular functions.

**c** Venn diagram of the non-overlapped biological effect associated with hypothermia-ROS synergistically induced cell death and overlapped biological effect associated with bacteria membrane induced immune response. **d** Histograms of differential proteins related to heat shock, oxidative stress, cell death, inflammatory response, bacteria infection, and immune response according to the GO annotation (*n* = 3 mice). **e** Differentially expressed protein interaction networks. The circle represents the gene and the line represents the protein-protein interaction. Data are presented as mean values ± SD. Source data are provided as a Source Data file.

additive or synergistic effects for amplified antitumor immune responses. For instance, biological membrane-based nanotechnology has shown enormous potential in fabricating therapeutic platforms for cancer treatment[10,42–44]. Nevertheless, the complicated procedures seriously impede the possibility of large-scale manufacturing. Different from conventional biological membrane-based nanoplatforms, in the present study, we showed that the electrochemically active *Shewanella oneidensis* MR-1 bacteria could be applied to produce FDA-approved PB MOFs on a large-scale taking advantage of its biological precipitation capability during the self-respiration process.

We presented that the resulting MiBaMc, consisting of PB MOF decorated bacteria membrane fragments having further modifications with photosensitive drug chlorin e6 and mitochondrial targeting moiety triphenylphosphine could specifically target mitochondria, thus amplifying the photo damages and finally inducing immunogenic cell death (Fig. 3). We also showed that, in mouse models with both immunogenic triple-negative breast tumor and colorectal tumor, MiBaMc triggered phototherapy combined with aPDL1 blocking antibody could effectively inhibit tumor growth. The tumor inhibition rates for triple-negative breast tumors and colorectal tumors were calculated to be 95.5% and 89.4%, respectively (Fig. 4b, f). The released tumor-associated antigens subsequently promote the maturation of dendritic cells in the tumor-draining lymph nodes, thus eliciting T cell-mediated immune response (Fig. 5). Given the increasing interest in cell membrane-based nanomedicine eliciting antitumor immune responses, it is conceivable that in situ biological precipitation of targeted nanoparticles using metabolic reductive bacteria is a promising strategy to enrich cell membrane-based nanoplatforms and cooperate with immune therapeutics. Notably, the major components (PB, Ce6, and aPDL1) within the nanoagent are FDA-approved, thus ensuring high biosafety and translation potential in the clinic[6,45].

At the same time, we also recognized the complexity of such an approach due to the mismatched absorption spectra of PB and Ce6, which may prolong treatment times. Thus, single laser/light-triggered simultaneous PTT and PDT using a dual-modal agent is preferred. In addition, the delivery of therapeutic light is crucial for successful clinical translation, especially for large and/or deep-seated tumors. In this case, one possible solution is the utilization of multiple interstitial fibers with decreased size and costs. Moreover, targeting the accumulation of therapeutic agents within the tumor tissues and even subcellular organelles, such as mitochondria, lysosomes, and nuclei, is also essential to enhance the photo-triggered cytotoxic activity.

In summary, we have developed a facile and efficient strategy to construct a bacteria membrane-Prussian blue hybrid nanoplatform in large-scale manufacturing. During the self-respiration process, uniform biologically precipitated Prussian blue MOFs were fabricated with *S. oneidensis* MR-1 bacteria as a supporting matrix to produce bacteria–PB hybrids. With fragmentation treatment and further modification with Ce6 and TPP, the resulting MiBaMc significantly localizes within the mitochondria and induces apoptosis of cancer cells via hyperthermia and ROS generation. After tumoral accumulation and endocytosis of MiBaMc, both hyperthermia and ROS are generated to induce the ICD of cancer cells under external light irradiation. When combined with aPDL1, MiBaMc shows potent tumor inhibition toward both immunogenic triple-negative breast cancer and colorectal cancer

models. This strategy paves the possible way for the development of multifunctional nanoplatforms using biological precipitation of targeted nanoparticles for cancer treatment.

## Methods

### Materials, cell lines, and animals

Potassium hexacyanoferrate ($K_3[Fe^{III}(CN)_6]$), ferric citrate hydrate ($C_6H_5Fe^{III}O_7$), sodium lactate ($CH_3CH(OH)COONa$), (3-carboxypropyl) triphenylphosphonium bromide (TPP-COOH), N-Hydroxysuccinimide (NHS), N-(3-dimethylaminopropyl)-N′-ethylcarbodiimide hydrochloride (EDC·HCl), and sodium lactate were purchased from Sigma-Aldrich. Photosensitizer chlorin e6 (Ce6) was obtained from Frontier Scientific, Inc. (Salt Lake City, UT, USA). Phosphate-buffered saline (PBS) and other chemicals were obtained from Thermo Fisher Scientific Pte. Ltd.

Mouse mammary carcinoma cell 4T1, murine colon adenocarcinoma cell MC38, and bacterial strains *S. oneidensis* (wide type, MR-1) were obtained from the American Type Culture Collection (ATCC). 4T1, MC38, and HEK293 cells were cultured at 37 °C in Roswell Park Memorial Institute (RPMI) 1640 and Dulbecco's modified Eagle's medium (DMEM) containing 10% fetal bovine serum (FBS) and 1% antibiotics (penicillin/streptomycin) in a humidified incubator filled with 5% $CO_2$. The bacterial contamination, fungi contamination, and mycoplasma contamination were all tested to be negative for all used cell lines. Balb/c female mice (6–8 weeks) were purchased from InVivos Pte. Ltd. (Singapore). All animal experiments were reviewed and approved with the Guidelines for Care and Use of Laboratory Animals of the Institutional Animal Care and Use Committee of Nanyang Technological University (NTU-IACUC) with a protocol number of A19016. Mice were housed in a temperature-constant animal room (22 °C) with reversed dark/light cycle (7:00 a.m. on and 7:00 p.m. off) and 40–70% humidity. The maximal tumor size/burden of 20 mm was permitted by the ethics committees and the maximal tumor size/burden in this study was not exceeded.

### Materials characterization

TEM images were acquired using a JEM-1400 transmission electron microscope (JEOL, Japan). The HAADF-STEM and EDS element mapping images were executed on a Talos F200X scanning transmission electron microscope (Thermo Fisher Corporation). PXRD patterns were acquired on a MiniFlex300/600 X-ray Diffractometer (Rigaku, Japan). SEM images were measured on a JSM-7600F Schottky field emission scanning electron microscope equipped with energy-dispersive X-ray (EDX) spectroscopy (JEOL, Japan). The Fe content was measured by an iCAP6000 inductively coupled plasma-optical emission spectrometer (Thermo Fisher Corporation). UV–vis–NIR absorption spectra were measured on a Shimadzu UV-3600 spectrometer (Japan). GIWAXS measurements were obtained from a Xenocs Nanoinxider with Cu-Kα microsource (40 nm) at 30 W. DLS tests were performed on a Malvern Nano-ZS ZEN3600 Particle Sizer (Malvern Instruments, UK). Flow cytometry was performed with Fortessa X20 (BD Biosciences) and analyzed by FlowJo v10. For the 660 nm LED light source, the central wavelength of the light is 660 nm with a full width at half maxima (FWHM) of 20 nm. For the 808 nm laser source, the output power was adjusted

according to the parameters provided by the manufacturer. Same light parameters were applied for in vitro and in vivo studies. CLSM images were performed on an LSM 800 confocal laser scanning microscope (ZEISS, Germany). Temperature measurement and in vivo thermal imaging were conducted on a FLIR thermal camera (FLIR E6, US). In vivo fluorescence imaging was performed on the IVIS fluorescence imaging system and analyzed by Living Image 4.3 software (IVIS-CT machine, PerkinElmer).

## Bacterial culture

*S. oneidensis* MR-1 bacteria were incubated in an LB medium at 30 °C with shaking for 24 h. After that, the bacteria were concentrated by centrifugation at $4000 \times g$ for 5 min and washed twice with buffer (0.5% NaCl solution). Finally, the washed *S. oneidensis* MR-1 cells were resuspended into a certain amount of buffer and bubbled with nitrogen gas to remove the dissolved oxygen. The final concentration of bacteria was about $1.5 \times 10^{10}$ CFU, which was tested by UV–vis absorption spectroscopy at 600 nm.

## Synthesis of S. oneidensis-MOFs

Sodium lactate (6 g, 60%, w/w) was added into NaCl solution (1 L, 0.5%) and bubbled with nitrogen to form a reaction solution. Then, potassium ferricyanide (0.5 M, 2.5 mL) and ferric citrate (0.25 M, 6.25 mL) were added to the anaerobic reaction solution (240 mL), followed by injecting bacterial dispersion (40 mL). After shaking for 24 h under 37 °C, the *S. oneidensis*-MOFs were collected by centrifugation ($4000 \times g$, 5 min) and washed with distilled water and saline solution. The final products were dispersed into a saline solution (40 mL) for storage. The large-scale synthesis of *S. oneidensis*-MOFs was realized by enlarging the volume of both the culture medium and the amount of bacteria.

## Synthesis of BaM

Storage solution (2 mL) of *S. oneidensis*-MOFs was mixed with $H_2O$ (3 mL) into a vial (10 mL in volume) with an ice bath, followed by sonicating via ultrasonic cell disruptor for 0.5, 1, and 2 h. The BaM with enough fragmentation (2 h sonication) was separated by centrifugation ($16,099 \times g$, 20 min) and washed several times with saline to ensure the removal of endolysates and secretions. The final products were redispersed into saline solution (1 mL) for storage.

## Electrochemical analysis

The electrochemical measurements were conducted in a three-electrode cell which consists of a working electrode (GC, 5 mm in diameter), a counter electrode (platinum plate), and a reference electrode (saturated calomel electrode) using a Reference 600 electrochemical workstation (Gamry). Concentrated bacterial dispersion (10 μL) was dropped onto the surface of the GC electrode and dried at room temperature for 5 min. Then, the cyclic voltammograms and long-term chronoamperometric measurements were conducted, respectively.

## Synthesis of BaMc and MiBaMc

For the synthesis of BaMc, Ce6 (6.0 mg), NHS (5.0 mg), and EDC·HCl (7.5 mg) were added to DMSO (200 μL) for stirring 2 h under dark conditions. The above solution was added dropwise into BaM (5 mL, 2 mg mL$^{-1}$) solution under vigorous stirring for 1 h. For the synthesis of MiBaMc, Ce6 (6.0 mg), TPP-COOH (4.0 mg), NHS (5 mg), and EDC·HCl (7.5 mg) were added into DMSO (200 μL) and stirred for 2 h under dark conditions. The above solution was added dropwise into BaM solution (5 mL, 2 mg mL$^{-1}$) under vigorous stirring for 1 h. The resulting BaMc and MiBaMc were separated by centrifugation ($16,099 \times g$, 20 min), washed thrice with PBS, and stored in a 4 °C fridge for further use.

## Photothermal effects

The photothermal effects of Prussian blue and MiBaMc and the same concentration of Prussian blue component (100 μg mL$^{-1}$) were performed under the irradiation of 808 nm laser with a power density of 0.5 W cm$^{-2}$. PBS buffer solution was set as the control. The temperatures at determined time points were recorded using a digital infrared thermal imager.

## Intracellular uptake of BaMc and MiBaMc

Murine breast 4T1 cancerous cells were incubated with BaMc and MiBaMc at a Ce6 concentration of 1.0 μg mL$^{-1}$ for different periods (1, 4, and 8 h). After incubation, the cells were washed with PBS and stained with MitoTracker™ Green FM and Hoechst 33342 (nuclear staining) for 30 min. After washing out free dyes, time-dependent intracellular localization of BaMc and MiBaMc was carried out on Carl Zeiss LSM 800 confocal laser microscope (Germany) and analyzed with ZEN 2012 software.

## Mitochondrial membrane potential

4T1 cancerous cells were treated with PBS (control), 808 nm (laser light source, 0.5 W cm$^{-2}$, 5 min) + 660 nm (LED light source, 10 mW cm$^{-2}$, 10 min) irradiation (NIR), BaM with 808 nm irradiation (BaM + PTT), BaMc with 660 nm irradiation (BaMc + PDT), BaMc with 808 nm + 660 nm irradiation (BaMc + PTDT), and MiBaMc with 808 nm + 660 nm irradiation (MiBaMc + PTDT). The fluence of 660 nm LED irradiation within 10 min was 6 J/cm$^2$. The fluence of 808 nm laser within 5 min irradiation was 150 J/cm$^2$. After irradiation, the cells were incubated with a culture medium containing 1X JC-10 dye working solution for 30 min. The cells were washed with PBS and visualized using CLSM (ZEISS, LSM800). The green and red signals detected represented JC-10 monomers and JC-10 aggregates, respectively. Besides, the quantitative analysis of JC-10 monomers and JC-10 aggregates was performed with flow cytometry (BD Biosciences, Fortessa X20).

## In vitro cytotoxicity

The biocompatibilities of Prussian blue MOFs (0, 6.25, 12.5, 25, 50, 100, 150, and 200 μg mL$^{-1}$) and MiBaMc (0, 6.25, 12.5, 25, 50, 100, 150, and 200 μg mL$^{-1}$) towards 4T1 cells were evaluated using standard methylthiazolyldiphenyl-tetrazolium bromide (MTT) assay. For in vitro killing tests, $1 \times 10^4$ 4T1 cells were seeded into 96-well plates. At an 80% confluency, cells were treated with different formulations (BaM + PTT, BaMc + PDT, BaMc + PTDT, and MiBaMc + PTDT with series concentrations (0, 1, 2, 4, and 8 μg mL$^{-1}$) based on Ce6. After treatment, the relative cell viability was evaluated by MTT assay.

## Intracellular ROS detection

4T1 cells were seeded into 8-well CLSM dishes and cultured overnight. Then cells were treated with different formulations (PBS, NIR, BaM + PTT, BaMc + PDT, BaMc + PTDT, and MiBaMc + PTDT). The equivalent concentration of Ce6 was 4 μg mL$^{-1}$. After irradiation, the cells were stained with a fluorescent ROS probe (DCFH-DA) for 2 h. The fluorescence imaging was performed with CLSM.

## Live/Dead staining and apoptosis analysis

4T1 cells were seeded into an 8-well CLSM dish and cultured overnight. After that, cells were treated with different formulations (PBS, NIR, BaM + PTT, BaMc + PDT, BaMc + PTDT, and MiBaMc + PTDT) for 24 h. The cells were stained with Calcein-AM and propidium iodide (PI) according to the manufacturer's protocols before conducting CLSM. For apoptosis analysis, 4T1 cells were seeded into six-well plates with different formulations. Then the total cells were collected for flow cytometry-based Annexin V-FITC/PI assays to determine the percentage of apoptosis.

## In vitro detection of ICD markers

The expressions of CRT on the 4T1 cancer cells membrane and intracellular expression of HMGB1 were determined using immunofluorescence assay and western blotting. 4T1 cells were first treated with different formulations and then washed with ice-cold PBS twice, fixed with 4% paraformaldehyde (PFA) for 30 min, and permeated with 0.1% PBST for 30 min at room temperature. For CRT, after blocking with 3% FBS, cells were stained with mouse anti-CRT (1:200) primary antibody at 4 °C overnight, followed by AF488-conjugated goat anti-mouse secondary antibody (dilution of 1:400). For HMGB1, after blocking with 3% FBS, cells were stained with rabbit anti-HMGB1 (1:400) primary antibody at 4 °C overnight, followed by AF647-conjugated goat anti-rabbit secondary antibody (dilution of 1:400). After washing out the unattached antibodies, the nuclei of cells were stained with Hoechst 33342 for 5 min and imaged under a CLSM. For western blotting, 4T1 cells treated with different formulations were lysed and heated at 95 °C for 5 min with sample loading buffer. After the purification by 4-12% gradient polyacrylamide gel electrophoresis, the products were transferred onto an Immuno-Blot PVDF membrane (Bio-Rad) and blocked with 3% BSA at room temperature. After that, target proteins were stained with mouse anti-CRT, rabbit anti-HMGB1, and rabbit anti-β-actin (1:1000) primary antibodies followed by HRP-labeled secondary antibodies (dilution of 1:10,000–1:20,000). The expression levels of target proteins based on enhanced chemiluminescence were monitored using an imaging system (ImageQuant 800).

## In vitro DCs maturation

To study DC maturation in vitro, bone marrow-derived DCs were collected from femurs of 4–6 weeks Balb/c female mice[46]. Typically, the bone marrow was collected and treated with red blood lysis buffer before culturing in a medium with granulocyte-macrophage colony-stimulating factor (GM-CSF, 20 ng mL$^{-1}$) and IL-4 (10 ng mL$^{-1}$) at 37 °C. The immature DCs were collected on day 7 for further experiments. 4T1 cells ($8 \times 10^5$ cells per well) were seeded into the upper chambers of transwells and pretreated with different formulations for 12 h before irradiation. Then, the immature DCs ($8 \times 10^5$ cells per well) were seeded into the lower chambers of the above transwells. After incubation for 12 h, the percentages of matured DCs in the lower chambers were determined by flow cytometry after staining with APC-conjugated anti-CD11c (dilution of 1:80), FITC-conjugated anti-CD80 (dilution of 1:50), and PE-conjugated anti-CD86 (dilution of 1:20) antibodies.

## In vitro cytokines release

$2 \times 10^5$ 4T1 cells were seeded into each well of six-well plates and pretreated with different formulations (PBS, NIR, BaM + PTT, BaMc + PDT, BaMc + PTDT, and MiBaMc + PTDT) and incubated for another 12 h. The released HMGB1 in the supernatants of 4T1 cells were collected and measured by an ELISA kit (FineTest) according to the protocol provided by the manufacturer. The secretion of ATP was also quantified with a luminescent ATP detection kit (Abcam) according to the protocol.

## Hemolysis analysis

Whole blood was collected from healthy Balb/c mice. Typically, red blood cells (RBCs) were collected with centrifugation at $4000 \times g$ for 20 min at 4 °C. The RBCs were further washed with fresh PBS several times until the supernatant was colorless. Next, RBCs suspended in PBS solution (0.5 mL) were mixed with PBS solution (0.5 mL) containing 2000, 1000, 800, 400, 200, 100, 50, and 25 μg mL$^{-1}$ of MiBaMc. Distilled water and PBS buffer were set as the positive control (100% hemolysis) and negative control (0% hemolysis), respectively. The supernatant of each concentration was collected with centrifugation after 2 h incubation at room temperature. The absorbance of the supernatant was measured with a Shimadzu UV-3600 spectrometer at 541 nm. The hemolysis rate (HR%) was calculated according

to following equation: HR% = [(Ab$_{sample}$ − Ab$_{negative\ control}$)/(Ab$_{positive\ control}$ − Ab$_{negative\ control}$)]×100%.

## In vivo fluorescence imaging

4T1 tumor model was established with 6−8 weeks Balb/c female mice. Typically, $1.5 \times 10^6$ cells 4T1 cells suspended in 40 μL PBS were subcutaneously injected into the right flank of mice. 7 days later, mice were intravenously administrated with free Ce6 and MiBaMc (equivalent dose based on Ce6: 4.0 mg kg$^{-1}$). Whole-body fluorescence imaging was conducted at pre-determined intervals with an IVIS fluorescence imaging system with excitation/emission wavelength at 675/703 nm. To study the biodistribution of free Ce6 and MiBaMc, the main organs and tumors were collected 24 h post-administration for ex vivo fluorescence imaging. At the same time, the amounts of Fe element originating from Prussian blue in main organs and tumors were quantified by inductively coupled plasma-optical emission spectroscopy (ICP-OES, Thermo Fisher).

## In vivo antitumor efficiency

Balb/c female mice (6−8 weeks) were injected with $1.0 \times 10^6$ 4T1 cells subcutaneously into the right flank. When the tumor volume reached 60 mm$^3$, mice were randomly divided into 9 groups ($n = 5$), including PBS, aPDL1, free Ce6 + PDT, BaMc + PDT, BaMc + PTT, BaMc + PTDT, MiBaMc + PTDT, BaMc + PTDT + aPDL1, and MiBaMc + PTDT + aPDL1. The dose and frequency for systematic administration of aPDL1 were 25 μg every day for 3 days through tail vein injection. All formulations were intravenously administrated the day after aPDL1 administration and followed by NIR irradiation 12 h later. Mouse weights and tumor volumes were monitored every two days with a calliper for 18 days. For the MC38 tumor model, mice were randomly divided into three groups ($n = 5$), including PBS, aPDL1, and MiBaMc + PTDT + aPDL1, and received the same treatment procedures. Tumor volumes were calculated according to the below equation: Volume = (Lt) × (Wt)$^2$ × 0.5 (mm$^3$), Lt is the length of the tumor, and Wt is the width of the tumor. And the relative tumor volume was determined as $V/V_0$, where $V_0$ represents the initial tumor volume. Tumors were collected and weighed at the endpoint of the treatment.

## H&E and TUNEL staining

After different treatments, the tumors and main organs of mice were surgically resected and fixed with 4% paraformaldehyde solution, followed by dehydration in gradient ethanol of 95%, 90%, 80%, and 70%. Those paraffin-embedded tissue blocks were further sectioned and stained using hematoxylin and eosin (H&E). The tumor tissue blocks were conducted with terminal deoxynucleotidyl transferase (TdT)-mediated deoxyuridine triphosphate (dUTP) nick end labeling (TUNEL) staining. Hoechst 33342 and FITC-labeled TUNEL assay kits were used to stain nuclei and DNA fragmentation, respectively. In apoptotic cells, FITC-labeled TdT catalyzed the incorporation of deoxynucleotides at the free 3'-hydroxyl ends of fragmented DNA. The stained tumor sections were visualized under a digital pathology system (Nikon, Eclipse C1).

## Blood biochemical analysis

Healthy Balb/c mice were intravenously administrated with MiBaMc. At determined time points post-administration, the whole blood and serum samples were collected for blood routine and liver and kidney indicators analysis.

## Flow cytometry analysis

Mice from each group were euthanized on day 13 post-inoculation. The draining lymph nodes, tumors, and spleens were collected. Those tissues were homogenized into single-cell suspension through mechanical mash or enzymatic digestion. After that, cells were filtered via 70 μm pore sterile cell strainers (Corning Inc.) and stained with

fluorescence-conjugated antibodies. For DCs maturation: FITC-conjugated anti-CD80 (Biolegend, cat no. 104705, dilution of 1:50), PE-conjugated anti-CD86 (Biolegend, cat no. 105007, dilution of 1:200), APC-conjugated anti-CD11c (Biolegend, cat no. 117309, dilution of 1:80), AF700-conjugated anti-CD45 (Biolegend, cat no. 103127, dilution of 1:200), and anti-CD16/32 (Biolegend, cat no. 156603, dilution of 1:200) antibodies were applied. For $CD3^+CD4^+$ and $CD3^+CD8^+$ T cells analysis: PE-conjugated anti-CD4 (Biolegend, cat no. 130310, dilution of 1:80), APC-conjugated anti-CD8a (Biolegend, cat no. 100711, dilution of 1:80), FITC-conjugated anti-CD3 (Biolegend, cat no. 100204, dilution of 1:50), AF700-conjugated anti-CD45 (Biolegend, cat no. 103127, dilution of 1:200), and anti-CD16/32 (Biolegend, cat no. 156603, dilution of 1:200) were used. For $CD3^+CD4^+Foxp3^+$ Treg analysis: cells were stained with anti-CD3, anti-CD4, and AF647-conjugated anti-Foxp3 (Biolegend, cat no. 126407, dilution of 1:100) antibodies. Compensation was carried out for T-cell analyses. All antibodies were used according to the protocols provided by the manufacturer. The stained cells were measured on a Fortessa X20 flow cytometer (BD Biosciences) and graphically analyzed using the FlowJo software package (version 10.0.7).

### Cytokine detection

Mouse serum samples from each treated group were collected on day 13 post-inoculation. The serum levels of TNF-α (Biolegend, cat. no. 430901), IFN-γ (Biolegend, cat. no. 4308.4), and IL-6 (Biolegend, cat. no. 431301) were measured with ELISA kits according to the protocols provided by the manufacturer.

### Immunofluorescence assay

Tumors were harvested from the mice in each group and frozen in the optimal cutting temperature medium. For CD4CD8 staining: tumors were cut via a cryotome, mounted on slides, and stained with AF488-conjugated anti-mouse CD8a antibody and AF647-conjugated anti-mouse CD4 antibody at 4 °C overnight. For HMGB1 and CRT staining, tumor tissues were first stained with rabbit anti-HMGB1 and mouse anti-CRT primary antibodies overnight at 4 °C, followed by AF647-conjugated goat anti-rabbit and AF488 conjugated goat anti-mouse secondary antibodies for another 2 h. The concentration of these antibodies used in the experiments was 20–40 µg/mL. Hoechst 33342 dye (Thermo-Fisher Scientific) was applied to stain cell nuclei. The slides were recorded using a confocal microscope (Nikon, Digital Eclipse C1 microscope system with NIKON DS-U3 controller). Hoechst 33342 was excited with a 405 nm laser, AF488 was excited with a 488 nm laser, and AF647 was excited with a 633 nm laser. The imaging channels were set at 450–500, 500–550, and 650–680 nm, respectively.

### Proteomics analysis

4T1 tumor-bearing BALB/c mice were randomly divided into two groups ($n = 3$). For MiBaMc + PTDT + aPDL1 group, the dose and frequency for systematic administration of aPDL1 were 25 µg every day for 3 days through tail vein injection. Later, mice in two groups were intravenously administrated with PBS or PBS containing MiBaMc (equivalent dose based on Ce6: 4.0 mg kg$^{-1}$). At 12 h post-administration, mice were treated with light irradiation. After different treatments, 4T1 tumor-bearing mice in each group ($n = 3$) were euthanized, and the tumor tissues were collected. The high-throughput sequencing and data analysis were performed in Oebitech Pte Ltd (China). The proteins were defined as differentially expressed if the fold-change of comparing the experimental groups and control groups was higher than 1.5 or lower than 0.67 with *P < 0.05. Protein–protein interaction network was figured out using the Search Tool for the Retrieval of Interacting Genes/Proteins (STRING) database (http://string-db.org/).

### Statistical analysis

Results were expressed as mean values ± SD unless otherwise stated. The statistical comparisons were conducted by one-way ANOVA with Tukey's tests or two-tailed unpaired Student's $t$-test. For all results, the values of $P < 0.05$ were considered to be statistically significant. Asterisk (*) denotes statistical significance between bars (****$P < 0.0001$, ***$P < 0.001$, **$P < 0.01$, and *$P < 0.05$). All statistical calculations were conducted using GraphPad Prism 8.0.

### Reporting summary

Further information on research design is available in the Nature Portfolio Reporting Summary linked to this article.

## Data availability

The proteomics data are available at iProX under Project IPX0005125000 [https://www.iprox.cn//page/project.html?id=IPX0005125000]. Source Data are provided with this paper. The authors declare that the remaining data supporting the findings of this study are available within the article, its Supplementary Information and Source Data file.

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

## Acknowledgements

This work was financially supported by the Singapore National Research Foundation under its Investigatorship (NRF-NRFI2018-03, Y.Z.) and Competitive Research Program (NRF-CRP26-2021-0002, Y.Z.). This work was also supported by the Fundamental Research Funds for the Central Universities (YD9990002021, D.W.).

## Author contributions

D.W., J.L., and Y.Z. conceived the idea and designed the project. D.W. and J.L. designed the experiments. D.W. and J.L. synthesized and characterized the materials and analyzed the results. D.W. performed the in vitro and in vivo experiments and analyzed the results. C.W. and Q.C. helped with the characterization of high-resolution TEM imaging and analyzed the results. W.Z., G.Y., Y.C., X.Z., Y.W., L.G., H.C., W.Y., and X.C. contributed to the experiment design and figure production. G.L. and B.G. conducted the electrochemical testing. D.W., J.L., and Y.Z. drafted the manuscript. All authors discussed the results and commented on the manuscript.

## Competing interests

The authors declare no competing interests.
