## [Peer Review File · Nature Communications]

Microbial synthesis of Prussian blue for potentiating checkpoint blockade immunotherapyReviewers' comments:

Reviewer #1 (Remarks to the Author):

Here, PB nanoparticles were synthesized taking advantage of bioprecipitation by *S. oneidensis* MR-1. After modification with Ce6 and TPP, the fragmented MiBaMc can target mitochondria for amplifying the combined photothermal and photodynamic damages and finally inducing ICD. Overall, this study demonstrated the potential of using biological precipitation of targeted nanoparticles for antitumor applications. In my opinion, this is a rigorous study with many characteristic data and sufficient discussion. Nevertheless, PB-relevant tumor therapies based on PTT or PDT and their immunogenicity effect have been well-reported thus far. Authors demonstrated the synthesis of PB via microbe. However, the unique advantages of the proposed approach in comparison with standard one-step PB synthetic method are not clear. Major concerns are put forward for authors' consideration.

1. The full name of "MiBaMc" need to be defined in the abstract and main text.
2. Authors believed that PB MOFs were generated on the surface of *S. oneidensis*. However, such MOF structure was not fully characterized. For example, the cubical crystal-like structure cannot be recognized from Figure 2a-d. The assemblies could also be the aggregates of PB salts.
3. For MiBaMc, it is not clear the weight ratio of PB, Ce6, TPP and bacteria membrane. In addition, the yield efficiency of PB on *S. oneidensis* need to be calculated.
4. In comparison with pristine PB, the MiBaMc may potentially cause severe immunoreaction in blood circulation, on account of the immunogenicity of bacteria membrane.
5. In Figure 3a, authors should first evaluate the Ce6 fluorescence quantum yield among free Ce6, BaMc and MiBaMc. High magnification image should be provided in Figure 3a.
6. The groups in Figure 3 need to be corrected. These four groups of "MiBaMc + PDT", "MiBaMc + PTT", "BaMc + PTDT" and "MiBaMc + PTDT" should be evaluated.
7. In Figure 3e, CRT and HMGB1 were co-stained with anti-HMGB1 primary antibody and anti-CRT antibody, followed by AF647-conjugated goat anti-rabbit secondary antibody. However, according to the ELISA principle, this will lead to an interference of antibody recognition. Moreover, HMGB1 requires permeabilization step, while CRT-staining does not need this step. Author should provide a detailed reason.
8. In Figure S9 and S10, the total number of cells is significantly varied in terms of different groups for flow cytometry and fluorescence microscopy. Therefore, such comparison would be invalid.
9. Why not show the photos of final xenograft tumor after various treatments? The liver function of mice is suggested to be evaluated when assessing the biosafety of MiBaMc.
10. The grouping of animal is not strictly consistent in Figure 5-6 and Figure S20-23.

Reviewer #2 (Remarks to the Author):

In this study Wang et al describe a photomedicine-based combination treatment strategy for synergistic enhancement of antitumor immunity. The approach entails a high degree of innovation, leveraging the self respiration process of a bacterium in the production of light-activated nanoparticles. The data presented is compelling, showing impressive tumor control in immunosuppressive mouse models treated with the authors MiBaMc combined with photothermal therapy, photodynamic therapy, and aPDL1 therapy. The results are supported by comprehensive mechanistic studies examining changes inflammatory cytokines and T cell populations in the various treatment groups. Overall the extent of mechanistic investigation and extent of resources leveraged to assess a wide range of biological endpoints is impressive. Of the concerns noted below questions about clinical feasibility are generally more significant than any issues surrounding technical rigour.

- 1) While the results achieved are impressive there is some concern that the complexity of this approach could be problematic for feasible clinical implementation. Even more straightforward implementation of phototherapies have taken decades to gain traction. To make the case for impact of this work it would help if a clear vision for clinical translation was presented. In this study preclinical models of breast and colon cancers are examined. But a clear case isn't presented for how this

complex treatment strategy - including consideration for therapeutic light delivery – would be achieved for these cancers.

2) Some important phototherapy dosimetry information is missing. For the mitochondrial membrane potential measurements it is specified that the irradiance at 660nm was 10mWcm^{-2} for a duration of ten minutes. However details are not provided about the light source (laser vs LED, spectral width etc) beam spot size, uniformity, and how power measurements were performed (make and model of power meter used etc). Also the fluence should be recorded ($600\text{ s} * 0.01\text{ W}^*\text{cm}^{-2} = 6\text{ J}^*\text{cm}^{-2}$). The power density is also specified for the photothermal light delivery but again details are missing about spot size, uniformity, light source and power measurements. Also, it isn't clear if the same parameters were applied for both in vitro and in vivo studies. Overall given the prominent role of phototherapy approaches here there should be much more detail provided about how these treatments were performed.

3) In Figure 6 the protein names and other labels are too small to be legible.

4) Awkward word choice in a few places detracts from clarity. The term “overlay” is used where perhaps “interaction” is intended?

Reviewer #3 (Remarks to the Author):

This manuscript by Wang and colleagues described a microbial synthesis of prussian blue and then used it as a reagent, in combination with anti-PD-L1, to treat mouse tumors. The microbial synthesis of prussian blue part is interesting with significant amount of in vitro data. However, the biology part is weak with some very confusing data and interpretation. Below are major concerns:

1) Fig 3i: After each treatment, 4T1 cells were labelled with CD80 and CD86, and the percentage of matured DCs was measured by flow cytometry. It is very confusing. Why were 4T1 cells labelled with CD80 and CD86? Did they actually measure CD80 and CD86 on DCs? If so the expression patterns of CD80 and CD86 are not convinced.

2) Some of in vivo data showed the favorable accumulation of MiBaMc in tumor. For example, for MiBaMc-treated mice, fluorescence signal within tumour region reached its maximum value at 12 h post-administration and maintained a relatively high fluorescence even for 24 h. However, it is unclear how MiBaMc would accumulate in tumor, as it was clear that MiBaMc could be internalized by many other cell types such as macrophages, DCs, etc. Sup Fig 13 showed significant fluorescence in non-tumor areas.

3) There was no difference among all tumor groups, even control group mice with large tumors had no body weight change, what did it mean?

4) In vivo mechanistic studies of photo-immunotherapy. Tumour-draining lymph nodes, spleen, and blood were studied, which is not the best approach to study tumor immunological responses. It would be more meaningful to study immune cells within the tumors.

5) Fig 5l showed CD8+cells (%) in G8 and G9 groups reached almost 100%, which is very confusing! If the results meant all these T cells were CD8+, the results would be strange.

6) Sup Fig 22. The FACS staining and/or getting on Foxp3 and CD4 were strange. Why would Foxp3+ cells be CD4- or CD4low cells?

Responses to Reviewer 1

*General comment: Here, PB nanoparticles were synthesized taking advantage of bioprecipitation by *S. oneidensis* MR-1. After modification with Ce6 and TPP, the fragmented MiBaMc can target mitochondria for amplifying the combined photothermal and photodynamic damages and finally inducing ICD. Overall, this study demonstrated the potential of using biological precipitation of targeted nanoparticles for antitumor applications. In my opinion, this is a rigorous study with many characteristic data and sufficient discussion. Nevertheless, PB-relevant tumor therapies based on PTT or PDT and their immunogenicity effect have been well-reported thus far. Authors demonstrated the synthesis of PB via microbe. However, the unique advantages of the proposed approach in comparison with standard one-step PB synthetic method are not clear. Major concerns are put forward for authors' consideration.*

Author's responses: We sincerely appreciate the respected reviewer for his/her recognition of our study as a rigorous study with many characteristic data and sufficient discussion. We also sincerely appreciate this reviewer for pointing out some drawbacks of our manuscript, especially the unique advantages of the biological precipitation strategy for Prussian blue (PB) synthesis. We have done our best to perform additional experiments to support our conclusions. We hope that the respected reviewer will be satisfied with our revisions.

It is known that the FDA has approved Prussian blue nanoparticles for safe and effective treatment of radioactive exposure in the clinic. Recent studies also confirmed its superior photothermal conversion ability, ensuring that it is a promising preclinical photothermal agent (*Adv. Mater.* 2020, 32, 2000542; *Nat. Commun.* 2019, 10, 4490)^{1,2}. Generally, two chemical strategies are employed for PB synthesis. One is a single-precursor strategy using either $K_4[Fe^{II}(CN)_6]$ or $K_3[Fe^{III}(CN)_6]$, and the other one is a co-precipitation based double-precursor strategy using equivalently mixed Fe^{III}/Fe^{II} and $[Fe^{II}(CN)_6]^{4-}/[Fe^{III}(CN)_6]^{3-}$ solution (*CrystEngComm* 2009, 11, 2257-2259; *J. Mater. Chem.* 2010, 20, 5251-5259)^{3,4}. However, the single-precursor strategy raises safety concerns about hydrogen cyanide generation, while the double-precursor strategy usually constrains dispersion uniformity and morphological regulation. In addition, chemical strategies for PB synthesis require relatively high reaction temperatures, harsh acidic environments, and additive surfactants to control the growth of PB nanocrystals and show low yield efficiency for PB (~21.5%). Moreover, chemical synthesis strategies also impose certain limitations for large-scale production as those methods involve dynamics, thermodynamics, and safety concerns. Thus, it is highly desired to develop facile strategies for controllable and large-scale synthesis of PB in a mild and eco-friendly manner.

Different from chemical synthesis strategies, biological precipitation of targeted nanoparticles using microorganisms (*e.g.*, bacteria, algae, fungi, and actinomycetes) has been regarded as a mild and eco-friendly strategy as it avoids the use of flammable or toxic chemicals. Especially, electrochemically active *S. oneidensis* MR-1 bacteria can be used as the

template to synthesize various metal nanoparticles (e.g., Pd, Pt, and Au) by reducing the corresponding high-valence metal salts on their membrane surface during the self-respiration process, which have been used in the field of catalysis, biochemical sensors, and biomedicine (*Sci. Adv.* 2016, 2, e1600858; *Science* 2016, 351, 74-77; *Sci. Adv.* 2020, 6, eaba1590; *Angew. Chem. Int. Ed.* 2011, 50, 427-430)⁵⁻⁸. Given the inspiration behind the utilization of Fe^{III} as an electron acceptor to support globally significant rates of respiration on early Earth, we speculated that *S. oneidensis* MR-1 bacteria could facilitate the biological reduction of Fe^{III} into Fe^{II} (*Science* 2015, 347, 1473-1476)⁹, which subsequently coordinated with [Fe^{III}(CN)₆] linkers to produce Prussian blue. In our study, the Fe^{III} ions from the starting ferric citrate (C₆H₅Fe^{III}O₇) were selectively reduced into Fe^{II} ions by the generated electrons on the membrane surface of *S. oneidensis* MR-1. Subsequently, the surrounding [Fe^{III}(CN)₆] linkers within potassium ferricyanide (K₃[Fe^{III}(CN)₆]) in the synthetic system would coordinate with Fe^{II} ions to form PB nanoparticles. Since Fe^{III} ions can only receive electrons on the surface of bacteria, such biological precipitation strategy can avoid the impacts of dynamics factors, which should be considered during the chemical synthesis strategy. The large-scale and mass production can be easily realized through expanding the number of bacteria and initial feeding of potassium ferricyanide (K₃[Fe^{III}(CN)₆]) and ferric citrate (C₆H₅Fe^{III}O₇). Moreover, the yield efficiency of FDA-approved PB with this biological precipitation strategy was calculated to be 43.4%, much higher than that of the chemical synthetic strategy.

In a very recent work (*Nat. Nanotechnol.* 2023, <https://doi.org/10.1038/s41565-023-01346-x>)¹⁰, authors constructed an artificial enzyme (MOF-derived Fe single-atomic nanozyme with both catalase and superoxide dismutase activities) modified *Bifidobacterium longum* probiotics for reshaping a healthy immune system in inflammatory bowel disease. Nevertheless, the synthesis of the microbe-nanoparticle system requires complicated procedures such as the modification of MOF-derived Fe single-atomic nanozyme, the conjugation of modified nanoparticles with the microbe, etc. Previous reports also showed the multienzyme-mimicking activity of PB MOFs, including peroxidase, catalase, and superoxide dismutase activity (*Adv. Mater.* 2022, 34, 2106723; *J. Am. Chem. Soc.* 2016, 138, 5860-5865)^{11,12}. Although the enzyme-mimicking activity of Prussian blue MOFs was not included in our present work, the bacteria-PB system holds a great potential in various biomedical fields such as neurodegenerative disease (Alzheimer's disease, Parkinson's disease), inflammatory-related disease (sepsis, inflammatory bowel disease, chronic wound for diabetic patients), and so on.

Cancer nanomedicine in combination with immunotherapies offers a great possibility to amplify antitumor immune responses and sensitize tumors to immunotherapies in a safe and effective manner. The natural world provides a host of materials and inspiration for the field of nanomedicine. Biological membrane-based nanotechnology has shown an enormous potential in fabricating platforms for drug delivery, immune manipulation, anti-bacteria, and theranostics (*Nature*, 2015, 526, 118; *Nat. Nanotechnol.* 2021, 16, 1271; *Nat. Nanotechnol.*

2022, 17, 531)¹³⁻¹⁵. While several strategies for cell membrane-originated vesicles have been reported, there are still some challenges, such as complicated procedures and large-scale manufacturing need to face in terms of clinical translation. Different from conventional cell membrane-based nanoparticles for theranostics, our study showed that electrochemically active *S. oneidensis* MR-1 bacteria could be applied to produce targeted nanoparticles (herein, FDA-approved Prussian blue nanoparticles) on a large-scale taking advantage of its biological precipitation capability during the self-respiration process. After ultrasonic fragmentation treatment, PB MOF decorated bacteria membrane fragments were obtained (named as BaM). We showed that the as-prepared MiBaMc, consisting of PB decorated bacteria membrane fragments (BaM) and further surface modification with photosensitive drug chlorin e6 and mitochondrial targeting moiety triphenylphosphine, could specifically target mitochondria. Notably, after fragmentation treatment, the systemic toxicity of the MiBaMc could be largely reduced as the use of intact bacteria poses safety concerns. After tumoral accumulation and endocytosis of MiBaMc, phototherapies induced hyperthermia and ROS were generated to induce immunogenic death (ICD) of tumor cells under light irradiation. The released tumor-associated antigens subsequently promoted the maturation of dendritic cells in the tumor-draining lymph nodes, thus eliciting T cell-mediated immune response. We also showed that in mouse models with both immunogenic triple-negative breast tumor and colorectal tumor, the combined phototherapies with aPDL1 blocking antibody could effectively inhibit tumor growth. The tumor inhibition rates for triple-negative breast tumor and colorectal tumor were calculated to be 95.5% and 89.4%, respectively. Plenty of experiments have also proven the antitumor efficiency and mechanism (e.g., western blot, flow cytometric analysis, proteomics analysis). The major components (Prussian blue, Ce6, and aPDL1) within MiBaMc are FDA-approved, thus ensuring high biosafety and high clinical translation potential. Given the increasing interest in cell membrane-based nanomedicine eliciting antitumor immune responses, it is conceivable that in situ biological precipitation of targeted nanoparticles using metabolic reductive bacteria might provide a new strategy to enrich cell membrane-based novel nanoplatforms and cooperate with immune therapeutics.

Comment 1: The full name of "MiBaMc" need to be defined in the abstract and main text.

Author's responses: Thanks for pointing out this flaw. We have defined MiBaMc in the abstract and main text. In our project, electrochemically active *S. oneidensis* MR-1 bacteria were first applied to synthesize PB MOFs on the surface (named as *S. oneidensis*-MOFs). After ultrasonic fragmentation treatment, PB MOF decorated bacteria membrane fragments were obtained (named as BaM). The final mitochondria-targeting therapeutic nanoplatform, namely MiBaMc, consists of PB MOF decorated bacteria membrane fragments (BaM) and further surface modification with photosensitizer Ce6 and mitochondria-targeting moiety triphenylphosphine (TPP) (Please check Page 1 and Page 2).

Comment 2: Authors believed that PB MOFs were generated on the surface of *S. oneidensis*. However, such MOF structure was not fully characterized. For example, the cubical crystal-like structure cannot be recognized from Figure 2a-d. The assemblies could also be the aggregates of PB salts.

Author's responses: In our project, the initial color of the starting system (*S. oneidensis* MR-1 bacteria + potassium ferricyanide ($K_3[Fe^{III}(CN)_6]$) + ferric citrate ($C_6H_5Fe^{III}O_7$)) was bright yellow (Fig. 2k). After biological precipitation by *S. oneidensis* MR-1 bacteria at 37 °C for 24 h, the final color changed into blue (a characteristic color of PB MOFs), indicating the successful synthesis of PB MOFs. The final products were then washed several times with distilled water and saline solution, and thus the existence of starting compounds potassium ferricyanide and ferric citrate could be excluded. In addition, powder XRD and grazing-incidence wide-angle X-ray scattering (GIWAXS) results showed the characteristic peaks from PB MOFs in the BaM, demonstrating the successful synthesis of PB MOFs. UV-vis-NIR spectra also showed a prominent absorption peak from PB MOFs. Overall, these results demonstrated the successful synthesis of PB nanocrystals via the biological precipitation of *S. oneidensis* MR-1 bacteria. Fig. 2a-d present the intact bacteria (*S. oneidensis* MR-1, light pink color), intact bacteria-PB MOFs (*S. oneidensis*-MOFs, blue color), BaM (blue color), and MiBaMc (dark blue color), respectively. To better show the cubical crystal-like PB MOFs on bacteria, we have revised Fig. 2a-d accordingly (Please check Page 4, Fig. 2a-d).

Fig. 2 | a-d, TEM images of (a) *S. oneidensis* MR-1 and (b) *S. oneidensis*-MOFs. Scale bar, 500 nm. TEM images of (c) BaM and (d) MiBaMc. Scale bar, 200 nm.

Comment 3: For MiBaMc, it is not clear the weight ratio of PB, Ce6, TPP and bacteria membrane. In addition, the yield efficiency of PB on *S. oneidensis* need to be calculated.

Author's responses: Thanks very much for this very important comment! As suggested, the weight ratio of PB, Ce6, TPP, and bacteria membrane within MiBaMc was calculated. Firstly, the weight of the PB component within 10 mg BaM was calculated to be 2.06 mg based on IPC-AES measurement. Thus, the PB and bacteria membrane weight ratio within BaM was 2.06/7.94. After that, 10 mg BaM was used for conjugation, and the Ce6 and TPP were measured to be 2.23 mg and 1.38 mg, respectively. Thus, the weight ratio of PB, Ce6, TPP, and bacteria membrane is 2.06/2.23/1.38/7.94. These data have been added to the revised manuscript (Please check Page 3).

In a typical procedure, *S. oneidensis* MR-1 bacteria (6.0×10^{11} CFU) were applied for PB

synthesis under anaerobic conditions. The feeding amounts of potassium ferricyanide ($K_3[Fe^{III}(CN)_6]$) and ferric citrate ($C_6H_5Fe^{III}O_7$) were 1.25 mmol and 1.5625 mmol, respectively. After shaking for 24 h under 37 °C, the as-prepared *S. oneidensis*-MOFs were collected by centrifugation and washed with distilled water and saline solution. Based on the ICP-AES results, the amount of synthetic PB on *S. oneidensis* MR-1 was 83.2 mg, and PB yield efficiency was calculated to be 43.4%. As a comparison, the chemical synthesis strategy (80 °C reaction temperature, acidic reaction solution with a pH value of 2, and surfactant protection) of PB was carried out based on a previous report (*Angew. Chem. Int. Ed.* 2012, 51, 984-988)¹⁶. As a result, 48.5 mg PB nanoparticles could be obtained with 226 mg precursor, and the calculated yield efficiency was 21.5%, much lower than the biological precipitation strategy. These data have been added to the revised manuscript (Please check Page 3).

Comment 4: In comparison with pristine PB, the MiBaMc may potentially cause severe immunoreaction in blood circulation, on account of the immunogenicity of bacteria membrane.

Author's responses: We greatly thank the reviewer for raising this professional comment! As suggested, we have performed the blood routine analysis of mice to demonstrate the possible inflammatory response induced by MiBaMc. Our results showed a slight increase in WBC, lymphocytes, monocytes, and neutrophils on day 1 post-administration of MiBaMc (24.4 mg/kg) compared to the control group (PBS). However, the levels of these indicators gradually restored to the normal range on day 3 and day 7 post-administration (Supplementary Figs. 21-24). These results demonstrated that MiBaMc could induce a low-level immune response in a very short period while did not cause long-term harm (Please check Page 7 and Page 8 for the discussion, and Supplementary Figs. 21-24 in the Supplementary Information).

Numbering: Control		Time: 20221221		Model: Whole blood	
Project	Result	Unit	Reference range		
WBC	5.0	10 ⁹ /L	0.8-6.8		
Lymph#	2.9	10 ⁹ /L	0.7-5.7		
Mon#	0.2	10 ⁹ /L	0.0-0.3		
Gran#	1.7	10 ⁹ /L	0.1-1.8		
Lymph%	57.9	%	55.8-90.6		
Mon%	4.8	%	1.8-6.0		
Gran%	37.3	%	8.6-38.9		
RBC	6.72	10 ¹² /L	6.36-9.42		
HGB	106	g/L	110-143		
HCT	34.2	%	34.6-44.6		
MCV	50.9	fL	48.2-58.3		
MCH	15.7	pg	15.8-19		
MCHC	309	g/L	302-353		
RDW	12.2	%	13-17		
PLT	458	10 ⁹ /L	450-1590		
MPV	7.3	fL	3.8-6.0		
PDW	17.1				
PCT	0.370	%			

Supplementary Fig. 21. Blood routine analysis. Whole blood test of mouse in control group injected with PBS.

Numbering: Day 1		Time: 20221221	Model: Whole blood	
Project	Result	Unit	Reference range	
WBC	8.8	10 ⁹ /L	0.8-6.8	
Lymph#	6.2	10 ⁹ /L	0.7-5.7	
Mon#	0.4	10 ⁹ /L	0.0-0.3	
Gran#	2.2	10 ⁹ /L	0.1-1.8	
Lymph%	70.4	%	55.8-90.6	
Mon%	4.2	%	1.8-6.0	
Gran%	25.4	%	8.6-38.9	
RBC	9.52	10 ¹² /L	6.36-9.42	
HGB	144	g/L	110-143	
HCT	46.4	%	34.6-44.6	
MCV	48.8	fL	48.2-58.3	
MCH	15.1	pg	15.8-19	
MCHC	310	g/L	302-353	
RDW	14.0	%	13-17	
PLT	808	10 ⁹ /L	450-1590	
MPV	6.7	fL	3.8-6.0	
PDW	16.5			
PCT	0.541	%		

Supplementary Fig. 22. Blood routine analysis. Whole blood test of mouse on day 1 post-injection with MiBaMc.

Numbering: Day 3		Time: 20221221	Model: Whole blood	
Project	Result	Unit	Reference range	
WBC	4.6	10 ⁹ /L	0.8-6.8	
Lymph#	2.7	10 ⁹ /L	0.7-5.7	
Mon#	0.2	10 ⁹ /L	0.0-0.3	
Gran#	1.7	10 ⁹ /L	0.1-1.8	
Lymph%	57.8	%	55.8-90.6	
Mon%	5.4	%	1.8-6.0	
Gran%	36.8	%	8.6-38.9	
RBC	8.64	10 ¹² /L	6.36-9.42	
HGB	132	g/L	110-143	
HCT	43.3	%	34.6-44.6	
MCV	50.2	fL	48.2-58.3	
MCH	15.4	pg	15.8-19	
MCHC	303	g/L	302-353	
RDW	14.5	%	13-17	
PLT	678	10 ⁹ /L	450-1590	
MPV	7.3	fL	3.8-6.0	
PDW	17.2			
PCT	0.202	%		

Supplementary Fig. 23. Blood routine analysis. Whole blood test of mouse on day 3 post-injection with MiBaMc.

Project	Result	Unit	Reference range
WBC	5.1	10 ⁹ /L	0.8-6.8
Lymph#	3.8	10 ⁹ /L	0.7-5.7
Mon#	0.2	10 ⁹ /L	0.0-0.3
Gran#	1.5	10 ⁹ /L	0.1-1.8
Lymph%	55.3	%	55.8-90.6
Mon%	4.7	%	1.8-6.0
Gran%	35.4	%	8.6-38.9
RBC	8.53	10 ¹² /L	6.36-9.42
HGB	126	g/L	110-143
HCT	41.0	%	34.6-44.6
MCV	51.4	fL	48.2-58.3
MCH	15.9	pg	15.8-19
MCHC	309	g/L	302-353
RDW	13.2	%	13-17
PLT	489	10 ⁹ /L	450-1590
MPV	7.0	fL	3.8-6.0
PDW	17.1		
PCT	0.342	%	

Supplementary Fig. 24. Blood routine analysis. Whole blood test of mouse on day 7 post-injection with MiBaMc.

Comment 5: In Figure 3a, authors should first evaluate the Ce6 fluorescence quantum yield among free Ce6, BaMc and MiBaMc. High magnification image should be provided in Figure 3a.

Author's responses: The fluorescence quantum yield efficiency of pure Ce6 in PBS was measured to be 0.17, 0.16, and 0.163 for pure Ce6, BaMc, and MiBaMc, respectively. Also, we have provided high-magnification images for the CLSM. Compared with free Ce6, both BaMc and MiBaMc showed an enhanced red fluorescence originating from Ce6 due to the enhanced endocytosis of nanoparticles. As shown in Fig. 3a, a punctate green fluorescence pattern from Mito-tracker can be witnessed. The overlap between the red fluorescence from MiBaMc was much higher than BaMc and free Ce6, indicating an enhanced localization within mitochondria for MiBaMc. Moreover, the Pearson's co-localization coefficients between the red Ce6 channel and the green mitochondria channel for free Ce6, BaMc, and MiBaMc were calculated to be 0.18 ± 0.08 , 0.29 ± 0.10 , and 0.51 ± 0.15 , respectively. Overall, these results demonstrated the enhanced Ce6 delivery efficacy and mitochondria-targeting ability of MiBaMc. (Please check Page 3 for the description, and Fig. 3a).

Fig. 3 | a, CLSM images showing the intracellular uptake of free Ce6, BaMc, and MiBaMc in 4T1 cancer cells after 8 h incubation. Blue, nucleus stained with Hoechst 33342; red, Ce6 fluorescence; green, mitochondria stained with Mito-Tracker green. Scale bar, 25 μm .

Comment 6: The groups in Figure 3 need to be corrected. These four groups of "MiBaMc + PDT", "MiBaMc + PTT", "BaMc + PTDT" and "MiBaMc + PTDT" should be evaluated.

Author's responses: We sincerely thank the reviewer for this comment! The "BaMc + PTDT" and "MiBaMc + PTDT" aim to confirm the mitochondria-targeting ability of MiBaMc after TPP modification. The two groups have already been included in our manuscript. The "MiBaMc + PDT", "MiBaMc + PTT", and "MiBaMc + PTDT" were shown to demonstrate that the therapeutic efficacy of the combined PTT and PDT (PTDT) is better than PDT and single PTT alone. To confirm this, we have performed the "BaM + PTT", "BaMc + PTT", and "BaMc + PTDT". Herein, we used "BaM + PTT" for the photothermal effect to avoid the possible interference from Ce6 as BaM only contains bacteria membrane and photothermal agent PB. We sincerely think the performed six groups can demonstrate the proposed concerns raised by the reviewer: Firstly, the combined photothermal and photodynamic therapy (PTDT) shows better therapeutic efficacy than PTT and single PDT alone; Secondly, the "MiBaMc + PTDT" group with mitochondrial targeting ability originated from TPP modification shows better therapeutic efficacy than the "BaMc + PTDT" group. Furthermore, we have strengthened the explanation in our revised manuscript (for Fig. 3) for better clarity.

Comment 7: In Figure 3e, CRT and HMGB1 were co-stained with anti-HMGB1 primary antibody and anti-CRT antibody, followed by AF647-conjugated goat anti-rabbit secondary antibody. However, according to the ELISA principle, this will lead to an interference of antibody recognition. Moreover, HMGB1 requires permeabilization step, while CRT-staining does not need this step. Author should provide a detailed reason.

Author's responses: We greatly appreciate the reviewer for this professional comment! To

avoid the possible interference between CRT and HMGB1, we have performed the in vitro CRT and HMGB1 staining individually. Typically, 4T1 cells were washed with ice-cold PBS twice after treatment with different formulations, then fixed with 4% paraformaldehyde (PFA) for 30 min, and permeated with 0.1% PBST for 30 min at room temperature. For CRT staining, after blocking with 3% FBS, cells were first stained with mouse anti-CRT (1:200) primary antibody at 4 °C overnight and washed three times with PBS, followed by AF488-conjugated goat anti-mouse secondary antibody (dilution of 1:400). For HMGB1 staining, after blocking with 3% FBS, cells were stained with rabbit anti-HMGB1 (1:400) primary antibody at 4 °C overnight and washed three times with PBS, followed by AF647-conjugated goat anti-rabbit secondary antibody (dilution of 1:400). After washing out the unattached secondary antibody, the nuclei of cells were stained with Hoechst 33342 for 5 min and imaged under CLSM. As shown in Fig. 3e, MiBaMc + PTDT group induced the most significant CRT exposure on the cell surface. Besides, the HMGB1 was located predominantly in the nuclei of 4T1 cells and gradually released from the nuclei after BaMc + PTDT treatment (Fig. 3f). The Figures have been added in the revised manuscript (Please check **Fig. 3**).

Fig. 3 | e,f, CLSM images showing CRT exposure (e) and HMGB1 release (f) in 4T1 cells after each treatment, respectively. Blue, nucleus stained with Hoechst 33342; Green, CRT stained with mouse anti-CRT primary antibody and then with Alexa Flour-488 conjugated goat anti-mouse secondary antibody; Red, HMGB1 stained with rabbit anti-HMGB1 primary antibody and then with Alexa Flour-647 conjugated goat anti-rabbit secondary antibody. Scale bar, 25 μm .

Comment 8: In Figure S9 and S10, the total number of cells is significantly varied in terms of different groups for flow cytometry and fluorescence microscopy. Therefore, such comparison would be invalid.

Author's responses: Thanks very much for this important comment! We have performed the flow cytometry and dead-live staining again. We carefully performed the washing procedure during the experiments, as some dead cells are easily washed away. For the flow cytometry analysis, the numbers of the cells between different groups are kept the same (Supplementary Fig. 9). For CLSM imaging of dead-live staining, some dead cells were easily removed during the washing process in our previous data, and thus the number of cells varied. Herein, we also

repeated the dead-live staining (Supplementary Fig. 10). The washing procedures were carefully performed to ensure the dead cells were still on the plates (Please check **Supplementary Figs. 9-10 in the Supplementary Information**).

Supplementary Fig. 9. Apoptosis assay 24 h after different treatments. **a**, 4T1 cancer cells treated with PBS, NIR irradiation, BaM + PTT, BaMc + PDT, BaMc + PTDT, and MiBaMc + PTDT at a Ce6 concentration of $4 \mu\text{g mL}^{-1}$. Cancer cells from different groups were co-stained with Annexin-V FITC and propidium iodide (PI) before performing Flow cytometry. **b**, Different cell populations obtained from (a) apoptosis assay.

Supplementary Fig. 10. Dead/Live staining using confocal laser scanning microscope. Confocal images of 4T1 cancer cells treated with PBS, NIR irradiation, BaM + PTT, BaMc + PDT, BaMc + PTDT, and MiBaMc + PTDT at a Ce6 concentration of $4 \mu\text{g mL}^{-1}$. The live cells and dead cells after calcein-AM/PI co-staining were displayed in green and red color, respectively. Scale bar, $50 \mu\text{m}$.

Comment 9: Why not show the photos of final xenograft tumor after various treatments? The liver function of mice is suggested to be evaluated when assessing the biosafety of MiBaMc.

Author's responses: We sincerely appreciate the reviewer for raising this valuable suggestion! For the 4T1 tumor-bearing mice model, we did not take the picture as the tumor tissues were

immediately applied for histological analysis and immunofluorescence studies. However, representative photos of tumor-bearing mice in each were recorded during the whole treatment (Supplementary Fig. 18). For the MC38 tumor-bearing mouse model, the initial purpose of this experiment was to demonstrate the universality of our therapeutic platform. Thus, we did not perform histological, immunofluorescence, or FCM analyses. A photo of the final xenograft MC38 tumors was provided (Supplementary Fig. 19).

As for your professional suggestion, we have performed the liver and kidney function analysis to study the long-term biocompatibility of MiBaMc. Typically, healthy balb/c mice were intravenously injected with MiBaMc, and the serum samples were collected for liver and kidney function analysis on day 7 or day 15 post-administration. As shown in Supplementary Fig. 20, liver functional indexes (ALP, ALT, AST) and kidney functional biomarkers (BUN and Crea) in the MiBaMc group were similar to those in the control group and remained within the normal ranges, further ensuring the biosafety of MiBaMc after systematic administration (Please check **Supplementary Figs. 18-20 in the Supplementary Information**).

Supplementary Fig. 18. Photos of mice for 4T1 tumor-bearing mice. Representative photo of 4T1 tumor-bearing mouse from each group at different time points during the treatment.

Supplementary Fig. 19. Photo of xenograft tumors from MC38 tumor-bearing mice. Photo of xenograft tumors after different treatments.

Supplementary Fig. 20. Liver and kidney function analysis of Balb/c mice at day 7 and day 15 post-administration of MiBaMc. **a**, Blood biochemistry assays of liver function markers for alkaline phosphatase (ALP), alanine transaminase (ALT), and aspartate aminotransferase (AST). **b**, Blood biochemistry assays of kidney function markers for blood urea nitrogen (BUN) and creatinine (Crea). Data were shown as mean \pm s.d. (n = 3 biologically independent mice per group).

Comment 10: The grouping of animal is not strictly consistent in Figure 5-6 and Figure S20-23.

Author's responses: We thank the respected reviewer for this important comment! For in vitro antitumor studies (Fig. 3), we have demonstrated that MiBaMc-triggered phototherapy (PTDT) showed enhanced therapeutic efficacy compared to PTT and PDT alone. For in vivo studies, 4T1 tumor-bearing mice were divided into 9 groups (G1: Control, G2: aPDL1, G3: free Ce6 + PDT, G4: BaMc + PDT, G5: BaMc + PTT, G6: BaMc + PTDT, G7: MiBaMc + PTDT, G8: BaMc + PTDT + aPDL1, and G9: MiBaMc + PTDT + aPDL1). Results showed that MiBaMc-triggered phototherapy (PTDT) combined with aPDL1 showed the most significant tumor therapeutic ability (Fig. 4). Thus, six representative groups (Control group, aPDL1, BaMc + PTDT, MiBaMc + PTDT, BaMc + PTDT + aPDL1, and MiBaMc + PTDT + aPDL1) were applied to study the in vivo immune responses and antitumor mechanism of MiBaMc triggered phototherapy (PTDT) combined with aPDL1 (Fig. 5). Such a grouping design could fully demonstrate our two conjectures: Firstly, stronger immune responses could be obtained for MiBaMc-treated groups than BaMc-treated groups on account of the mitochondria-targeting ability of MiBaMc. Secondly, MiBaMc-triggered phototherapy combined with clinically approved aPDL1 checkpoint blockade could further elicit the immune responders for enhanced antitumor efficacy. To keep the grouping consistent, we have added the immunofluorescence staining analysis of HMGB1, CRT, and CD4⁺/CD8⁺ T cells, as well as ROS staining of tumor sections for aPDL1, BaMc + PTDT, and MiBaMc + PTDT groups (Please check **Fig. 5** and **Supplementary Fig. 31 in the Supplementary Information**).

While for previous Figure S21, the data showed the secretions of proinflammatory cytokines (IFN- γ , TNF- α , and IL-6) in peripheral blood serum from a more immunosuppressive MC38 colorectal tumor model. The grouping design was based on the in vivo therapy studies

of the MC38 tumor model (Fig. 4f-h), which was independent of the 4T1 tumor model. Herein, a more immunosuppressive MC38 colorectal tumor model was applied to demonstrate the universality of our therapeutic platform.

Fig. 5 | i, Immunofluorescence staining of HMGB1 in nuclei (upper), CRT (middle), and CD4⁺ and CD8⁺ T cells (lower) in tumor sections after treatment of each formulation. Nuclei (blue) were stained by Hoechst 33342. Scale bar, 100 μm.

Supplementary Fig. 31. ROS fluorescence images of tumor slices. Representative ROS fluorescence images of tumor sections 12 h after different treatments. Green fluorescence signal represents ROS level. For ROS staining of tumor tissues: the mice were first intravenous administration of PBS and MiBaMc, 12 h later the mice were intraperitoneal administration of DCFH-DA (5 mg/kg), followed by irradiation or not. The tumors were surgically excised, and frozen tumor slices were prepared. Hoechst 33342 dye was applied to stain cell nuclei. Scale bar, 1000 μm. The fluorescent tumor slides were recorded using a confocal microscope (Nikon, Digital Eclipse C1 microscope system with NIKON DS-U3 controller).

Responses to Reviewer 2

General comment: *In this study Wang et al describe a photomedicine-based combination treatment strategy for synergistic enhancement of antitumor immunity. The approach entails a high degree of innovation, leveraging the self respiration process of a bacterium in the production of lightactivated nanoparticles. The data presented is compelling, showing impressive tumor control in immunosuppressive mouse models treated with the authors MiBaMc combined with photothermal therapy, photodynamic therapy, and aPDL1 therapy. The results are supported by comprehensive mechanistic studies examining changes inflammatory cytokines and T cell populations in the various treatment groups. Overall the extent of mechanistic investigation and extent of resources leveraged to assess a wide range of biological endpoints is impressive. Of the concerns noted below questions about clinical feasibility are generally more significant than any issues surrounding technical rigour.*

Author's responses: We really appreciate the respected reviewer for raising these valuable suggestions! We are very grateful for the precious time and superb professionalism the reviewer has put into improving our work. We sincerely hope the respected reviewer will be satisfied with our revisions. Thank you so much for your positive suggestion!

Comment 1: While the results achieved are impressive there is some concern that the complexity of this approach could be problematic for feasible clinical implementation. Even more straightforward implementation of phototherapies have taken decades to gain traction. To make the case for impact of this work it would help if a clear vision for clinical translation was presented. In this study preclinical models of breast and colon cancers are examined. But a clear case isn't presented for how this complex treatment strategy - including consideration for therapeutic light delivery - would be achieved for these cancers.

Author's responses: This is a very professional and constructive comment! As we know, photodynamic therapy (PDT) induces the death of tumor cells through the localized activation of photosensitizers within tumors. It has been used in the clinic for more than 40 years for the treatment of various cancers, including superficial skin lesions and oesophageal and lung tumors (*Photochem. Photobiol.* 2020, 96, 506-516; *J. Clin. Oncol.* 2017, 35, e14056; *Lasers Surg. Med.* 2006, 38, 445-467).¹⁷⁻¹⁹ The PDT agents include redaporfin (a bacteriochlorin), hemoporphin (haematoporphyrin monomethyl ether), photocyanine (a zinc phthalocyanine), photodithazine (a chlorin e6 derivative), radachlorin (a chlorin and purpurin mixture), chlorin e6 sodium-polyvinylpyrrolidone (photolon; a chlorin e6 derivative formulation), and so on. Several of them have reached the clinical stages of development, and some have achieved clinical approval in China and Russia. Especially, chlorin e6 is a second-generation, FDA-approved photosensitizer (FDA UNII: 5S2CCF3T1Z) with improved pharmacokinetic properties and has been approved as N-aspartyl chlorin e6 in Japan to treat lung cancer (*Mater.* 2013, 6, 817-840)²⁰. Although photothermal therapy (PTT) agents have not yet been tested in large

clinical trials, laser-induced ablation without PTT agents (laser device only) has already been used clinically, as thermal ablation can be achieved via excitation of endogenous chromophores within human tissues. Also, several medical devices that do not require contrast agents have been successfully developed for cancer PTT (*Radiology* 2002, 225, 367-377; *Radiology* 2004, 230, 450-458)²¹⁻²². Contrast-enhanced PTT is an emerging focus of research and can largely improve the selectivity for target tissue and reduce the requirement of high laser power. For example, contrast-enhanced PTT that uses gold nanoshells for prostate cancer ablation is now an ongoing clinical trial (NCT04240639). The combination of PDT and PTT shows a synergistic effect as the heating effect of PTT can improve local blood flow and increase the oxygen concentration within tumor tissues, thus showing an enhanced PDT efficacy. In turn, the ROS generated during PDT can disrupt heat-shock proteins, thereby negating their protective effects in tumor cells during PTT. Additionally, the combination of PDT and PTT with other therapeutic modalities could provide opportunities to exploit the advantages while overcoming the shortcomings of each therapeutic modality. For example, PDT (Redaporfin) has been combined with immune-checkpoint inhibitors (anti-PD1 antibodies) in the clinical setting, with a sustained complete response in a patient with Head and Neck squamous cell carcinoma (HNSCC) refractory to multiple prior therapies (*Case Rep. Oncol.* 2018, 11, 769-776).²³ These results demonstrate the significance and potential clinical translation of combined therapy in cancer treatment, especially the combination of PTT, PDT, and immunotherapy.

In our case, electrochemically active *S. oneidensis* MR-1 bacteria were used to produce FDA-approved Prussian blue (PB) MOFs on a large-scale, taking advantage of its biological precipitation capability during the self-respiration process. After sonication, PB MOF decorated bacteria membrane fragments were obtained (named as BaM). The final mitochondria-targeting nanoplatfrom, namely MiBaMc, consists of PB MOF decorated bacteria membrane fragments and further surface modification with photosensitizer Ce6 and mitochondria-targeting moiety triphenylphosphine. The FDA has already approved PB nanoparticles in the clinic for safe and effective treatment of radioactive exposure. Recent research also confirmed its superior photothermal conversion ability and PTT efficacy. Typically, for the in vivo therapeutic studies (breast and colon cancer models) in our research, the corresponding cancer cells were inoculated subcutaneously into the right flank of Balb/c mice, and the MiBaMc nanoparticles were intravenously injected. After tumoral accumulation and endocytosis of MiBaMc, hyperthermia (808 nm laser at a power density of 0.5 W/cm² for 5 min) and ROS (660 nm LED light at a power density of 10 mW/cm² for 10 min) were generated to induce immunogenic death (ICD) of tumor cells. It should also be noted that the irradiations were only applied to tumor sites, thus largely avoiding the possible damage to normal tissues. As a result, the MiBaMc achieved a satisfying tumor inhibition rate for both the 4T1 tumor model (95.5%) and a more immunosuppressive MC38 tumor model (89.4%). Notably, the major components (PB, Ce6, and aPDL1) within MiBaMc are FDA-approved, thus ensuring high

biosafety and high clinical translation potential.

At the same time, we also recognized the complexity of such an approach due to the mismatched absorption spectra of the PTT agent (808 nm) and PDT agent (660 nm), which will prolong treatment times. Thus, single laser/light triggered simultaneous PTT and PDT, based on the use of a PTT agent coupled with a PDT agent or a dual-modal photothermal and photodynamic agent is preferred, which may not be covered by this work, and we will continue our exploration in future studies. In addition, we totally agreed with the respected reviewer that the delivery of therapeutic light is crucial for successful clinical translation, especially for large and/or deep-seated tumors. In this case, one possible solution for photo-induced therapy for large and/or deep-seated tumors is the utilization of multiple interstitial fibers with decreased size and costs. Except for the agents and light delivery, the targeting accumulation of therapeutic agents within tumor tissues and even the subcellular organelles, such as mitochondria, lysosomes, and nuclei, is also essential to enhance the cytotoxic activity of PDT and PTT. In our case, mitochondria-targeting moiety triphenylphosphine, which preferentially inserts into the inner membrane of mitochondria, was applied onto the surface of MiBaMc to realize the active targeting ability in mitochondria and subsequently enhance the PDT killing efficacy. Overall, as preclinical interest in combined PTT and PDT therapy continues to increase, the effectiveness, ease of use, and competitiveness compared to well-established modalities should be considered as key metrics in these development efforts. Some of these discussions are included in the discussion part of our revised manuscript (Please check **the Discussion part on Pages 7-9**).

Comment 2: Some important phototherapy dosimetry information is missing. For the mitochondrial membrane potential measurements it is specified that the irradiance at 660 nm was 10 mWcm^{-2} for a duration of ten minutes. However details are not provided about the light source (laser vs LED, spectral width etc) beam spot size, uniformity, and how power measurements were performed (make and model of power meter used etc). Also the fluence should be recorded ($600 \text{ s} * 0.01 \text{ W} * \text{cm}^{-2} = 6 \text{ J} * \text{cm}^{-2}$). The power density is also specified for the photothermal light delivery but again details are missing about spot size, uniformity, light source and power measurements. Also, it isn't clear if the same parameters were applied for both in vitro and in vivo studies. Overall given the prominent role of phototherapy approaches here there should be much more detail provided about how these treatments were performed.

Author's responses: This is a very constructive and professional comment! The parameters of light sources are included in the revised manuscript as per the advice. For the 660 nm LED light source, the central wavelength of the light is 660 nm with a full width at half maxima (FWHM) of 20 nm. By adjusting the working current and distance between the light source, a $5 \text{ cm} * 6 \text{ cm}$ area could be obtained with a uniform power density of 10 mW/cm^2 . The power density of 660 nm LED was determined by an optical power meter (Q8230, ADVANTEST). The fluence

of 660 nm LED irradiation within 10 min was calculated to be $600 \text{ s} * 0.01 \text{ W/cm}^2 = 6 \text{ J/cm}^2$. For the 808 nm laser source, the output power was adjusted to 0.5 W, and the round spot size was adjusted to 1 cm^2 according to the parameters provided by the manufacturer, thus providing a power density of 0.5 W/cm^2 . The power stability of the 808 nm laser is $\pm 0.9\%$, according to the laser data report provided by the manufacturer. The fluence of 808 nm laser within 5 min irradiation was calculated to be $300 \text{ s} * 0.5 \text{ W/cm}^2 = 150 \text{ J/cm}^2$.

In our project, the same parameters were applied for both in vitro and in vivo studies (808 nm laser at a power density of 0.5 W/cm^2 for 5 min, 660 nm LED light at a power density of 10 mW/cm^2 for 10 min). For in vivo study, the mice were under anesthesia when applying the irradiations. For PTT, 808 nm laser irradiation with a spot size of 1 cm^2 was applied to the tumor area. For PDT, only the tumor tissues were exposed to 660 nm LED light irradiation. At the same time, the remaining parts of the body were covered with aluminum foil to minimize the possible damage to normal tissues. Those parameters are provided in the experimental section of the main manuscript and Supplementary Information (Please check the **Characterization part in the Supplementary Information**).

Comment 3: In Figure 6 the protein names and other labels are too small to be legible.

Author's responses: Thanks! We have revised the size of protein names and other labels from 4 to 6 pt (Please check **Fig. 6**).

Comment 4: Awkward word choice in a few places detracts from clarity. The term "overlay" is used where perhaps "interaction" is intended?

Author's responses: Thanks! The term "overlay" has been revised into "overlap" (Please check **Page 8 and Page 9**).

Responses to Reviewer 3

General comment: This manuscript by Wang and colleagues described a microbial synthesis of prussian blue and then used it as a reagent, in combination with anti-PD-L1, to treat mouse tumors. The microbial synthesis of prussian blue part is interesting with significant amount of in vitro data. However, the biology part is weak with some very confusing data and interpretation. Below are major concerns:

Author's responses: We really appreciate the reviewer's constructive, professional, and helpful comment! We are extremely grateful for the precious time and superb professionalism the respected reviewer has put into improving our manuscript's quality. We also deeply appreciate this reviewer for pointing out some drawbacks of our manuscript, especially for the in vivo characterizations of immune responses. We have done our best to perform additional experiments to support our conclusions accurately. We sincerely hope the respected reviewer will be satisfied with our revisions.

Comment 1: Fig 3i: After each treatment, 4T1 cells were labelled with CD80 and CD86, and the percentage of matured DCs was measured by flow cytometry. It is very confusing. Why were 4T1 cells labelled with CD80 and CD86? Did they actually measure CD80 and CD86 on DCs? If so the expression patterns of CD80 and CD86 are not convinced.

Author's responses: Thanks very much for this professional comment! On account of our carelessness, we here made a typo in the Figure caption of Fig. 3i! For the study of DC maturation in vitro, bone marrow-derived DCs were first collected from femurs of 4-6 weeks Balb/c female mice and treated with red blood lysis buffer before culturing in medium with granulocyte-macrophage colony-stimulating factor (GM-CSF, 20 ng mL⁻¹) and IL-4 (10 ng mL⁻¹) at 37 °C. After that, the immature DCs were collected on day 7 for further experiments. As shown in Supplementary Fig. 12, 4T1 cells (8 × 10⁵ cells per well) were seeded into the upper chambers of transwells and pretreated with different formulations for 12 h before irradiation. Then, the immature DCs (8 × 10⁵ cells per well) were seeded into the lower chambers of the above transwells. After incubation for 12 h, the maturation of DCs was determined by flow cytometry after staining with APC-conjugated anti-CD11c, FITC-conjugated anti-CD80, and PE-conjugated anti-CD86 antibodies. We have revised the figure caption of Fig. 3i accordingly. A schematic illustration for the ICD-induced DCs maturation was added in the Supplementary Information (Please check Page 5 for the discussion, Figure caption for Fig. 3, and Supplementary Fig. 12 in the Supplementary Information).

Supplementary Fig. 12. In vitro maturation of DCs triggered by MiBaMc. Schematic illustration of ICD-induced maturation of DCs in vitro.

Comment 2: Some of in vivo data showed the favorable accumulation of MiBaMc in tumor. For example, for MiBaMc-treated mice, fluorescence signal within tumour region reached its maximum value at 12 h post administration and maintained a relatively high fluorescence even for 24 h. However, it is unclear how MiBaMc would accumulate in tumor, as it was clear that MiBaMc could be internalized by many other cell types such as macrophages, DCs, etc. Sup Fig 13 showed significant fluorescence in non-tumor areas.

Author's responses: We thank the respected reviewer for this very important comment! Herein, MiBaMc was used as a nanoplatform. For nanoparticle-based platforms, the accumulation within the tumor site is based on the well-known enhanced permeability and retention (EPR) effect (*Adv. Drug Delivery Rev. 2011, 63, 136-151*).²⁴ The EPR effect is a unique phenomenon of solid tumors related to their anatomical and pathophysiological differences from normal tissues, which serves as a basis for the development of nanomedicine. Typically, angiogenesis leads to high vascular density in solid tumors, significant gaps exist between endothelial cells in tumor blood vessels, and tumor tissues exhibit selective extravasation and retention of nanoparticles. Thus, the enhanced tumor accumulation of MiBaMc should be attributed to the EPR effect, while free Ce6 molecule cannot. Undeniably, MiBaMc showed relative accumulation within normal tissues such as the liver, spleen, kidney. This is a universal phenomenon for nanomedicine on account of the existence of monocyte-macrophage system (MPS) during blood circulation, and such phenomenon has been observed in numerous studies (*Nat. Nanotechnol. 2021, 16, 1260-1270; Nat. Nanotechnol. 2021, 16, 1130-1140; Nat. Commun. 2021, 12, 523*)²⁵⁻²⁷. In our case, in vivo fluorescence imaging results showed the highest average radiant efficiency within tumor tissue compared to other normal tissues. While there may be some accumulation in other organs, the light irradiations were only applied locally within the tumor sites, thus largely avoiding the possible side-effects to normal

tissues, as verified by the H&E staining results of major organs (Supplementary Fig. 21). Furthermore, the in vivo biocompatibility of MiBaMc was studied. Typically, healthy balb/c mice were intravenously injected with MiBaMc, and the serum samples were collected for liver and kidney function analysis on day 7 or day 15 post-administration. As shown in Supplementary Fig. 20, liver functional indexes (ALP, ALT, AST) and kidney functional biomarkers (BUN and Crea) in the MiBaMc group were similar to those in the control group and remained within the normal ranges, further ensuring the biosafety of MiBaMc after systematic administration (Please check Page 6 for the discussion).

Supplementary Fig. 21. Photographs of H&E stained sections of main organs from 4T1 tumor-bearing mice after different treatments. Mice were treated with various formulations, and the main organs (heart, liver, spleen, lung, and kidney) were isolated after treatment for H&E staining (n = 3). Scale bar, 100 μ m.

Supplementary Fig. 20. Liver and kidney function analysis of Balb/c mice at day 7 and day 15 post-administration of MiBaMc. **a**, Blood biochemistry assays of liver function markers for alkaline phosphatase (ALP), alanine transaminase (ALT), and aspartate aminotransferase (AST). **b**, Blood biochemistry assays of kidney function markers for blood urea nitrogen (BUN) and creatinine (Crea). Data were shown as mean \pm s.d. (n = 3 biologically independent mice per group).

Comment 3: There was no difference among all tumor groups, even control group mice with large tumors had no body weight change, what did it mean?

Author's responses: We greatly thank this reviewer for raising this important comment! For the in vivo therapeutic study, we found that the mice in the control group showed low appetite and emaciated figures due to the increasing high burden of tumors over time. Thus, the mice in the control group are relatively thinner even with large tumors. At the same time, mice in experimental groups with smaller tumors and lower tumor burden were in good condition and had a good appetite. Thus, we did not observe too big differences between the average body weights among all groups. Other studies also showed a similar phenomenon (*Nat. Nanotechnol.* 2019, 14, 379-387; *Nat. Commun.* 2022, 13, 3468)²⁸⁻²⁹. In those studies, the experimental groups showed no apparent body weight compared to the control group. This may be due to the individual differences among each other of the experimental mice. Besides, we have modified the range of ordinates to make the differences between each group more obvious. As shown in Fig. 4e, as the mice were randomly divided into 9 groups, the differences between each group existed at the beginning as the mice were divided randomly based on the average tumor volume.

Comment 4: In vivo mechanistic studies of photo-immunotherapy. Tumour-draining lymph nodes, spleen, and blood were studied, which is not the best approach to study tumor immunological responses. It would be more meaningful to study immune cells within the tumors.

Author's responses: Thank you for this professional suggestion! Except for the immunological responses within tumor-draining nodes, spleens, and blood, we also performed the immunological responses within tumor tissues by checking the percentages of CD3⁺CD4⁺ helper T cells, CD3⁺CD8⁺ cytotoxic T cells, and CD3⁺CD4⁺Foxp3⁺ Tregs. It is well-known that both cytotoxic T lymphocytes (CTL) (CD3⁺CD8⁺ T cells) and helper T cells (CD3⁺CD4⁺ T cells) are critical to regulating adaptive immunities. As shown in Fig. 5c-e, the percentage of activated CD3⁺CD8⁺ T cells in the MiBaMc + PTDT + aPDL1 group (29.1%) was 1.22-, 1.46-, and 2.75-fold higher than BaMc + PTDT + aPDL1, MiBaMc + PTDT, and control groups, respectively. In addition to CTL recruitment, MiBaMc triggered phototherapy combined with aPDL1 also elicited the most significant infiltration of helper T cells (42.5%) in the tumor, showing 1.18-, 1.26-, and 2.31-fold increase in population compared to the BaMc + PTDT + aPDL1, MiBaMc + PTDT, and control groups, respectively. Furthermore, the expression of representative biomarkers of ICD (HMGB1 and CRT) within tumor sections was studied by immunofluorescence staining. Results showed significantly decreased HMGB1 expression within the nuclei and amplified CRT exposure in MiBaMc + PTDT + aPDL1 group (Fig. 5i). Besides, MiBaMc + PTDT + aPDL1 group showed the highest population of both CD3⁺CD4⁺ helper T cells and CD3⁺CD8⁺ cytotoxic T cells within the tumor sections, as verified by the

immunofluorescence staining. Moreover, the analysis and discussion of CD3⁺CD4⁺Foxp3⁺ Tregs within tumor tissues were also provided in response to **Comment 6** below. Overall, the above results indicated that the MiBaMc triggered phototherapy combined with aPDL1 could effectively elicit the immune responders, an essential step to mount an antitumor immunity. The above discussions and the figures have been added to the revised manuscript (Please check **Pages 8 and 9** for the discussion, and **Supplementary Fig. 30** in the **Supplementary Information**).

Fig. 5 | c, Representative flow cytometric plots of the T cells in 4T1-bearing tumor tissue gating on CD3⁺ cells after treatment of each formulation (n = 3 biologically independent mice). **d,e**, Quantitative analysis of the CD3⁺CD8⁺ cytotoxic T cells and CD3⁺CD4⁺ helper T cells as a percentage of CD3⁺ lymphocytes based on flow cytometric results (**c**). Statistical analysis was conducted by one-way ANOVA. n.s. represents none of significance, **P* < 0.05, ***P* < 0.01, ****P* < 0.001, *****P* < 0.0001.

Fig. 5 | i, Immunofluorescence staining of HMGB1 in nuclei (upper), CRT (middle), and CD4⁺ and CD8⁺ T cells (lower) in tumor sections after treatment of each formulation. Nuclei (blue) were stained by Hoechst 33342. Scale bar, 100 μm.

Comment 5: Fig 5I showed CD8+ cells (%) in G8 and G9 groups reached almost 100%, which is very confusing! If the results meant all these T cells were CD8+, the results would be strange.

Author's responses: Thanks for the comment! In our previous version, 100% (for the G9 group, MiBaMc + PTDT + aPDL1) was set as a normalized value for a more intuitive comparison. We have again performed the immunofluorescence staining of CD4+ and CD8+ T cells in tumor sections, and more groups were involved. In our revised manuscript, the figure was drawn with raw data. As shown in Supplementary Fig. 29, the number of CD8+ T cells per $100 \times 10^3 \mu\text{m}^2$ area was analyzed after different treatments. MiBaMc + PTDT + aPDL1 (G9) group showed the highest population of CD4+ helper T cells and CD8+ cytotoxic T cells within the tumor sections (Please check the **Supplementary Fig. 29 in the Supplementary Information**).

Supplementary Fig. 29. Number of CD8+ T cells in tumor sections of 4T1 tumor-bearing mice after each treatment. Quantitative analysis of CD8+ T cells in the sections of 4T1 tumor-bearing mice after each treatment. Data were shown as mean \pm s.d. (n = 6). Statistical analysis was conducted using one-way ANOVA. * $P < 0.05$, ** $P < 0.001$, **** $P < 0.0001$.

Comment 6: Sup Fig 22. The FACS staining and/or getting on Foxp3 and CD4 were strange. Why would Foxp3+ cells be CD4- or CD4 low cells?

Author's responses: We sincerely appreciate the reviewer for raising this very important comment! Regulatory T cells (Tregs, CD3+CD4+Foxp3+) are immunosuppressive and generally suppress or down-regulate the induction and proliferation of effector cells. As shown in Supplementary Fig. 30, we have re-performed the FCM analysis of Tregs within tumor tissues after different treatments. For the gating strategy, we added a "CD3+ vs. CD4+" gating after

"CD3⁺ vs. SSC-A", and the Foxp3⁺ cells were based on the CD3⁺CD4⁺ cells. The results indicated the populations of Tregs for MiBaMc + PTDT + aPDL1 and MiBaMc + PTDT dramatically decreased by 84.3% and 63.4% compared to the control group injected with PBS, respectively. (Please check **Supplementary Fig. 30 in the Supplementary Information**).

Supplementary Fig. 30. In vivo immune response in tumor tissues after different treatments. **a**, Gating strategy to analyze CD3⁺CD4⁺Foxp3⁺ Tregs. **b**, Population of CD3⁺CD4⁺Foxp3⁺ Tregs in the tumor tissues of each group a flow cytometric analysis according to flow cytometry. **c**, Quantitative analysis of the CD3⁺CD4⁺Foxp3⁺ Tregs in each group ($n = 3$ biologically independent mice per group). Statistical analysis was conducted using one-way ANOVA. n.s. represents none of significance, ** $P < 0.01$, **** $P < 0.0001$.

References

- [1] Wu, W.; Yu, L.; Pu, Y.; Yao, H.; Chen, Y.; Shi, J., Copper-Enriched Prussian Blue Nanomedicine for In Situ Disulfiram Toxicification and Photothermal Antitumor Amplification. *Adv. Mater.* **2020**, *32*, 2000542.
- [2] Li, J.; Liu, X.; Tan, L.; Cui, Z.; Yang, X.; Liang, Y.; Li, Z.; Zhu, S.; Zheng, Y.; Yeung, K. W. K.; Wang, X.; Wu, S., Zinc-doped Prussian blue enhances photothermal clearance of *Staphylococcus aureus* and promotes tissue repair in infected wounds. *Nat. Commun.* **2019**, *10*, 4490.
- [3] Hu, M.; Jiang, J.; Ji, R.; Zeng, Y., Prussian blue mesocrystals prepared by a facile hydrothermal method. *CrystEngComm* **2009**, *11*, 2257.
- [4] Shokouhimehr, M.; Soehnen, E. S.; Hao, J.; Griswold, M.; Flask, C.; Fan, X.; Basilion, J. P.; Basu, S.; Huang, S. D., Dual purpose Prussian blue nanoparticles for cellular imaging and drug delivery: a new generation of T₁-weighted MRI contrast and small molecule delivery agents. *J. Mater. Chem.* **2010**, *20*, 5251-5259.
- [5] Liu, J.; Zheng, Y.; Hong, Z.; Cai, K.; Zhao, F.; Han, H., Microbial synthesis of highly dispersed PdAu alloy for enhanced electrocatalysis. *Sci. Adv.* **2016**, *2*, e1600858.
- [6] Sakimoto, K. K.; Wong, A. B.; Yang, P., Self-photosensitization of nonphotosynthetic bacteria for solar-to-chemical production. *Science* **2016**, *351*, 74-77.
- [7] Dong, X.; Pan, P.; Zheng, D. W.; Bao, P.; Zeng, X.; Zhang, X. Z., Bioinorganic hybrid bacteriophage for modulation of intestinal microbiota to remodel tumor-immune microenvironment against colorectal cancer. *Sci. Adv.* **2020**, *6*, eaba1590.
- [8] Wu, X.; Zhao, F.; Rahunen, N.; Varcoe, J. R.; Avignone-Rossa, C.; Thumser, A. E.; Slade, R. C., A role for microbial palladium nanoparticles in extracellular electron transfer. *Angew. Chem. Int. Ed.* **2011**, *50*, 427-430.
- [9] Byrne, J. M.; Klueglein, N.; Pearce, C.; Rosso, K. M.; Appel, E.; Kappler, A., Redox cycling of Fe(II) and Fe(III) in magnetite by Fe-metabolizing bacteria. *Science* **2015**, *347*, 1473-1476.
- [10] Cao, F.; Jin, L.; Gao, Y.; Ding, Y.; Wen, H.; Qian, Z.; Zhang, C.; Hong, L.; Yang, H.; Zhang, J.; Tong, Z.; Wang, W.; Chen, X.; Mao, Z., Artificial-enzymes-armed *Bifidobacterium longum* probiotics for alleviating intestinal inflammation and microbiota dysbiosis. *Nat. Nanotechnol.* **2023**. doi:10.1038/s41565-023-01346-x.
- [11] Ma, X.; Hao, J.; Wu, J.; Li, Y.; Cai, X.; Zheng, Y., Prussian Blue Nanozyme as a Pyroptosis Inhibitor Alleviates Neurodegeneration. *Adv. Mater.* **2022**, *34*, 2106723.
- [12] Zhang, W.; Hu, S.; Yin, J. J.; He, W.; Lu, W.; Ma, M.; Gu, N.; Zhang, Y., Prussian Blue Nanoparticles as Multienzyme Mimetics and Reactive Oxygen Species Scavengers. *J. Am. Chem. Soc.* **2016**, *138*, 5860-5.
- [13] Hu, C. M.; Fang, R. H.; Wang, K. C.; Luk, B. T.; Thamphiwatana, S.; Dehaini, D.; Nguyen, P.; Angsantikul, P.; Wen, C. H.; Kroll, A. V.; Carpenter, C.; Ramesh, M.; Qu, V.; Patel, S. H.; Zhu, J.; Shi, W.; Hofman, F. M.; Chen, T. C.; Gao, W.; Zhang, K.; Chien, S.; Zhang, L., Nanoparticle biointerfacing by platelet membrane cloaking. *Nature* **2015**, *526*, 118-21.
- [14] Zhai, Y.; Wang, J.; Lang, T.; Kong, Y.; Rong, R.; Cai, Y.; Ran, W.; Xiong, F.; Zheng, C.; Wang, Y.;

- Yu, Y.; Zhu, H. H.; Zhang, P.; Li, Y., T lymphocyte membrane-decorated epigenetic nanoinducer of interferons for cancer immunotherapy. *Nat. Nanotechnol.* **2021**, *16*, 1271-1280.
- [15] Liu, C.; Liu, X.; Xiang, X.; Pang, X.; Chen, S.; Zhang, Y.; Ren, E.; Zhang, L.; Liu, X.; Lv, P.; Wang, X.; Luo, W.; Xia, N.; Chen, X.; Liu, G., A nanovaccine for antigen self-presentation and immunosuppression reversal as a personalized cancer immunotherapy strategy. *Nat. Nanotechnol.* **2022**, *17*, 531-540.
- [16] Hu, M.; Furukawa, S.; Ohtani, R.; Sukegawa, H.; Nemoto, Y.; Reboul, J.; Kitagawa, S.; Yamauchi, Y., Synthesis of Prussian blue nanoparticles with a hollow interior by controlled chemical etching. *Angew. Chem. Int. Ed.* **2012**, *51*, 984-988.
- [17] Hamblin, M. R., Photodynamic Therapy for Cancer: What's Past is Prologue. *Photochem. Photobiol.* **2020**, *96*, 506-516.
- [18] Oliveira, J.; Monteiro, E.; Santos, J.; Silva, J. D.; Almeida, L.; Santos, L. L., A first in human study using photodynamic therapy with Redaporfin in advanced head and neck cancer. *J. Clin. Oncol.* **2017**, *35*, e14056.
- [19] Pandey, R. K.; Goswami, L. N.; Chen, Y.; Gryshuk, A.; Missert, J. R.; Oseroff, A.; Dougherty, T. J., Nature: a rich source for developing multifunctional agents. tumor-imaging and photodynamic therapy. *Lasers Surg. Med.* **2006**, *38*, 445-467.
- [20] Ormond, A. B.; Freeman, H. S., Dye Sensitizers for Photodynamic Therapy. *Materials* **2013**, *6*, 817-840.
- [21] Vogl, T. J.; Straub, R.; Eichler, K.; Woitaschek, D.; Mack, M. G., Malignant liver tumors treated with MR imaging-guided laser-induced thermotherapy: experience with complications in 899 patients (2,520 lesions). *Radiology* **2002**, *225*, 367-377.
- [22] Vogl, T. J.; Straub, R.; Eichler, K.; Söllner, O.; Mack, M. G., Colorectal Carcinoma Metastases in Liver: Laser-induced Interstitial Thermotherapy-Local Tumor Control Rate and Survival Data. *Radiology* **2004**, *230*, 450-458.
- [23] Santos, L. L.; Oliveira, J.; Monteiro, E.; Santos, J.; Sarmiento, C., Treatment of Head and Neck Cancer with Photodynamic Therapy with Redaporfin: A Clinical Case Report. *Case Rep. Oncol.* **2018**, *11*, 769-776.
- [24] Fang, J.; Nakamura, H.; Maeda, H., The EPR effect: Unique features of tumor blood vessels for drug delivery, factors involved, and limitations and augmentation of the effect. *Adv. Drug Delivery Rev.* **2011**, *63*, 136-151.
- [25] Wang, Z.; Little, N.; Chen, J.; Lambesis, K. T.; Le, K. T.; Han, W.; Scott, A. J.; Lu, J., Immunogenic camptothosome nanovesicles comprising sphingomyelin-derived camptothecin bilayers for safe and synergistic cancer immunochemotherapy. *Nat. Nanotechnol.* **2021**, *16*, 1130-1140.
- [26] Sun, X.; Zhang, Y.; Li, J.; Park, K. S.; Han, K.; Zhou, X.; Xu, Y.; Nam, J.; Xu, J.; Shi, X.; Wei, L.; Lei, Y. L.; Moon, J. J., Amplifying STING activation by cyclic dinucleotide-manganese particles for local and systemic cancer metalloimmunotherapy. *Nat. Nanotechnol.* **2021**,

16, 1260-1270.

- [27] Tang, W.; Yang, Z.; He, L.; Deng, L.; Fathi, P.; Zhu, S.; Li, L.; Shen, B.; Wang, Z.; Jacobson, O.; Song, J.; Zou, J.; Hu, P.; Wang, M.; Mu, J.; Cheng, Y.; Ma, Y.; Tang, L.; Fan, W.; Chen, X., A hybrid semiconducting organosilica-based O₂ nanoeconomizer for on-demand synergistic photothermally boosted radiotherapy. *Nat. Commun.* **2021**, *12*, 523.
- [28] Gong, N.; Ma, X.; Ye, X.; Zhou, Q.; Chen, X.; Tan, X.; Yao, S.; Huo, S.; Zhang, T.; Chen, S.; Teng, X.; Hu, X.; Yu, J.; Gan, Y.; Jiang, H.; Li, J.; Liang, X. J., Carbon-dot-supported atomically dispersed gold as a mitochondrial oxidative stress amplifier for cancer treatment. *Nat. Nanotechnol.* **2019**, *14*, 379-387.
- [29] Zhang, C.; Huang, J.; Zeng, Z.; He, S.; Cheng, P.; Li, J.; Pu, K., Catalytic nano-immunocomplexes for remote-controlled sono-metabolic checkpoint trimodal cancer therapy. *Nat. Commun.* **2022**, *13*, 3468.

REVIEWERS' COMMENTS

Reviewer #1 (Remarks to the Author):

Authors' replies have addressed the concerns from the reviewer, and the supplementary data are satisfactory. Biological synthetic strategy is promising for formulating functional nanoparticles. The presented approach may inspire more facile nanoplatform designs. Thus, this manuscript could be considered for publication as is in this high-impact journal now.

Reviewer #2 (Remarks to the Author):

Concerns raised in the previous round of reviews have been adequately addressed here

Reviewer #3 (Remarks to the Author):

The authors addressed all of my questions.

Responses to Reviewer 1

Reviewer #1 (Remarks to Author):

Authors' replies have addressed the concerns from the reviewer, and the supplementary data are satisfactory. Biological synthetic strategy is promising for formulating functional nanoparticles. The presented approach may inspire more facile nanoplatform designs. Thus, this manuscript could be considered for publication as is in this high-impact journal now.

Author's responses: Thanks very much for your recommendation of publication.

Responses to Reviewer 2

Reviewer #2 (Remarks to Author):

Concerns raised in the previous round of reviews have been adequately addressed here

Author's responses: Thanks very much for your recommendation of publication.

Responses to Reviewer 3

Reviewer #3 (Remarks to Author):

The authors addressed all of my questions.

Author's responses: Thanks very much for your recommendation of publication.